**EMBO** *reports*

# ZBTB48 is both a vertebrate telomere-binding protein and a transcriptional activator

Arne Jahn[1,2,†] (ID), Grishma Rane[3,†] (ID), Maciej Paszkowski-Rogacz[1], Sergi Sayols[4], Alina Bluhm[4], Chung-Ting Han[4,‡] (ID), Irena Draškovič[5], José Arturo Londoño-Vallejo[5] (ID), Alan Prem Kumar[3,6,7], Frank Buchholz[1,8,9,10,11,*] (ID), Falk Butter[4,**] (ID) & Dennis Kappei[3,***] (ID)

## Abstract

Telomeres constitute the ends of linear chromosomes and together with the shelterin complex form a structure essential for genome maintenance and stability. In addition to the constitutive binding of the shelterin complex, other direct, yet more transient interactions are mediated by the CST complex and HOT1/HMBOX1, while subtelomeric variant repeats are recognized by NR2C/F transcription factors. Recently, the Kruppel-like zinc finger protein ZBTB48/HKR3/TZAP has been described as a novel telomere-associated factor in the vertebrate lineage. Here, we show that ZBTB48 binds directly both to telomeric and to subtelomeric variant repeat sequences. ZBTB48 is found at telomeres of human cancer cells regardless of the mode of telomere maintenance and it acts as a negative regulator of telomere length. In addition to its telomeric function, we demonstrate through a combination of RNAseq, ChIPseq and expression proteomics experiments that ZBTB48 acts as a transcriptional activator on a small set of target genes, including mitochondrial fission process 1 (MTFP1). This discovery places ZBTB48 at the interface of telomere length regulation, transcriptional control and mitochondrial metabolism.

**Keywords** gene regulation; mitochondria; telomere length; telomeres; transcription factor

**Subject Categories** Chromatin, Epigenetics, Genomics & Functional Genomics; Transcription

See also: **L Garcia-Exposito & RJ O'Sullivan** (June 2017)

## Introduction

Telomeres are the ends of linear chromosomes and in vertebrates are composed of DNA with the hexameric motif TTAGGG that are bound by a dedicated set of proteins. Their main functions are to prevent loss of genetic information due to replicative shortening and to prevent recognition of the ends of linear chromosomes from recognition as double-strand breaks. The latter is mediated by the six-protein shelterin complex that is constituently bound to telomeres and that contains the two DNA double-strand binders TRF1 and TRF2 and the single-strand binder POT1 [1]. In addition to the shelterin complex, additional direct telomere-binding factors that are more transiently interacting with telomeres or the shelterin complex also assist in telomere maintenance. For example, the CST complex binds to the single-stranded telomeric overhangs and this interaction is critical both for telomere replication and termination of telomerase activity [2,3]. Furthermore, NR2C/F transcription factors have been described to bind to subtelomeric variant repeats in ALT cells, to promote targeted telomere insertion (TTI) and more generally recombination events at telomeres in the interplay with the chromatin remodelling NuRD complex and the zinc finger protein ZNF827 [4,5].

Telomere shortening, which ultimately culminates in cellular senescence, is counteracted by the reverse transcriptase telomerase in stem cells and in about 85% of all cancers, while the remaining cancers maintain telomeres with the recombination-based

1   Medical Systems Biology, UCC, University Hospital and Medical Faculty Carl Gustav Carus, TU Dresden, Dresden, Germany
2   Institute for Clinical Genetics, Medical Faculty Carl Gustav Carus, TU Dresden, Dresden, Germany
3   Cancer Science Institute of Singapore, National University of Singapore, Singapore City, Singapore
4   Institute of Molecular Biology (IMB) gGmbH, Mainz, Germany
5   Telomeres & Cancer Laboratory, UMR3244, Institut Curie-CNRS-UPMC, Paris Cedex 05, France
6   Department of Pharmacology, Yong Loo Lin School of Medicine, National University of Singapore, Singapore City, Singapore
7   Curtin Medical School, Faculty of Health Sciences, Curtin University, Perth, Australia
8   Max Planck Institute of Molecular Cell Biology and Genetics, Dresden, Germany
9   German Cancer Research Center (DKFZ), Heidelberg, Germany
10  German Cancer Consortium (DKTK) Partner Site Dresden, Dresden, Germany
11  National Center for Tumor Diseases (NCT), University Hospital Carl Gustav Carus, TU Dresden, Dresden, Germany
    *Corresponding author. Tel: +49 351 46340288; E-mail: frank.buchholz@tu-dresden.de
    **Corresponding author. Tel: +49 6131 39 21570; E-mail: f.butter@imb-mainz.de
    ***Corresponding author. Tel: +65 6516 1333; E-mail: dennis.kappei@nus.edu.sg
    †These authors contributed equally to this work
    ‡Present address: CeGaT GmbH, Center for Genomics and Transcriptomics, Tübingen, Germany

alternative lengthening of telomeres (ALT) mechanism. Beyond these obvious implications for ageing and carcinogenesis, telomeres have been linked to a multitude of other diseases including cardiovascular and pulmonary diseases, neurodegeneration, autoimmune diseases, diabetes and obesity [6]. In particular, metabolic disorders tightly link telomeres to mitochondrial metabolism and dysfunctions in telomeres and telomeric proteins have been shown to compromise mitochondrial homeostasis and vice versa [7,8]. A thorough understanding of the molecular players at telomeres is thus critical for a wide variety of health conditions.

In a search for such factors, several large-scale screens have described hundreds of putative telomere-associated factors [9–13]. Among those, HOT1 (also known as HMBOX1) was identified both by proteomics of isolated chromatin segments (PICh) and an *in vitro* reconstitution DNA–protein interaction screen combined with quantitative, high-resolution mass spectrometry [9,10]. We have previously characterized HOT1 as a direct telomeric dsDNA-binding protein and as a positive regulator of telomere length contributing to telomerase recruitment [10]. The *in vitro* reconstitution approach has since then been extended to systematically investigate telomere-binding proteins in 16 vertebrate species, creating a phylointeractomics map of telomeres [13]. ZBTB48 (also known as HKR3 or TZAP [14]) is among the most conserved factors that were found to be associated with TTAGGG repeats. Here, we show that ZBTB48 is indeed a direct (sub)telomere-binding protein based on a zinc finger-TTAGGG interaction and acts as a negative regulator of telomere length as recently shown independently of our study [14]. Beyond its telomeric role, we further demonstrate that ZBTB48 also acts as a transcriptional activator, regulating the expression of a defined set of target genes. Among those, the expression of mitochondrial fission process 1, MTFP1, is dependent on ZBTB48, extending ZBTB48's role in telomere homeostasis to the integrity of the mitochondrial network.

# Results

### ZBTB48 binds to telomeric DNA through its zinc finger 11

The identification of ZBTB48 in our previous phylointeractomics screen in 16 different vertebrate species was due to its ability to associate with TTAGGG repeat sequences [13]. With 11 adjacent zinc fingers (ZnF) including one degenerated ZnF (ZnF2), ZBTB48 contains several putative DNA-binding domains. To test which ZnF is responsible for mediating telomere binding, we expressed FLAG-ZBTB48 WT and point mutants by exchanging the first histidine to alanine of the 10 functional $Cys_2His_2$ ZnFs in HeLa cells and performed DNA pull-downs using either telomeric DNA or a scrambled control as baits. In agreement with our previous identification, FLAG-ZBTB48 WT was strongly enriched on the telomeric but not on the control DNA (Fig 1A and B). While point mutants of ZnF1-10 maintained TTAGGG-binding ability, mutation of ZnF11 (ZBTB48 H596A, ZnF11mut) led to a complete loss of enrichment on telomeric DNA, which we further confirmed by a series of additional deletion constructs (Fig EV1A). To conversely test whether ZnF11 is sufficient for binding, we deleted ZnF1-10 from the FLAG-ZBTB48 construct. Indeed, FLAG-ZBTB48 ΔZnF1-10 efficiently bound to TTAGGG repeats (Figs 1A and B, and EV1A), showing that ZnF11 is both necessary and sufficient for telomere binding. To further

address the specificity of the TTAGGG recognition, we tested binding of FLAG-ZBTB48 WT to the most common subtelomeric variant repeat motifs TTGGGG, TCAGGG and TGAGGG [15,16]. Both TTGGGG and TCAGGG repeats were bound efficiently, while for TGAGGG only a weak enrichment was detected (Fig 1C). In all cases, no binding was detected with the FLAG-ZBTB48 ZnF11mut, again confirming its function to mediate binding to telomere-like sequences. Other variant sequences such as telomeric motifs found in *C. elegans* (TTAGGC) [17], *A. mellifera* (TTAGG) [18] and *T. castaneum* (TCAGG) [19] were not recognized by FLAG-ZBTB48 WT (Fig EV1B). These data demonstrate that ZBTB48 recognizes TTAGGG and subtelomeric variant repeats via its ZnF11. Thus, in contrast to TRF1, TRF2 and HOT1, which do not recognize subtelomeric variant repeats [10,20], the *in vitro* binding pattern of ZBTB48 is rather reminiscent of NR2C/F transcription factors [4,21].

### ZBTB48 binds to telomeres *in vivo*

To validate an *in vivo* association of ZBTB48 with telomeres, we performed co-localization experiments for ZBTB48 with TRF2, a constituent marker of telomeres, in unsynchronized U2OS cells. Using an antibody against endogenous ZBTB48, which we validated in U2OS ZBTB48 knockout (KO) cells (Fig EV2A–E), ZBTB48 showed a punctuated nuclear pattern that largely overlapped with telomeric foci marked by TRF2. On average, about half of all TRF2 signals co-localized with ZBTB48 (Fig 1D). To further validate that ZnF11 mediates the interaction with telomeric DNA, we expressed exogenous FLAG-ZBTB48 WT as well as point mutants for ZnF10 and ZnF11. While all three constructs showed a bright nuclear punctuated pattern, only FLAG-ZBTB48 WT and FLAG-ZBTB48 ZnF10mut co-localized with TRF2 (Fig 1E). In contrast, co-localization between TRF2 and FLAG-ZBTB48 ZnF11mut was reduced to background levels (Fig 1E). Compared to endogenous ZBTB48, FLAG-ZBTB48 WT overexpression led to an increased frequency of co-localization with TRF2, suggesting that ZBTB48 binding to telomeres is dose dependent and limited at least by its own expression level. Similarly, in U2OS the ZBTB48 protein shows frequent co-localization with PML bodies (Fig 1F), which in this ALT-positive cancer cell line are associated with telomeres and form ALT-associated PML bodies (APBs) [22]. Again, FLAG-ZBTB48 WT and FLAG-ZBTB48 ZnF10mut recapitulated the co-localization observed for endogenous ZBTB48 while expression of FLAG-ZBTB48 ZnF11mut showed a largely reduced co-localization frequency (Fig EV1D). Together these data indicate that ZBTB48 is a telomere-binding protein *in vitro* and *in vivo* and that ZnF11 mediates its binding to telomeric DNA.

In contrast to U2OS cells, we detected low frequencies of co-localization events between telomeres and endogenous ZBTB48 in telomerase-positive HeLa Kyoto cells with short telomeres, which we used for functional experiments throughout this study, with a moderate yet significant increase when overexpressing FLAG-ZBTB48 WT (Fig EV1E). To test whether this was simply due to the relatively short telomeres in these HeLa cells, we investigated ZBTB48 localization in HeLa 1.3 cells with long telomeres as well as in HT1080 super-telomerase cells, with all four cell lines used showing similar ZBTB48 protein expression levels (Fig EV1C). HT1080 super-telomerase cells are derived from telomerase-positive cells with short telomeres, but have engineered long telomeres due to concomitant overexpression of both TERT and TERC [23].

 

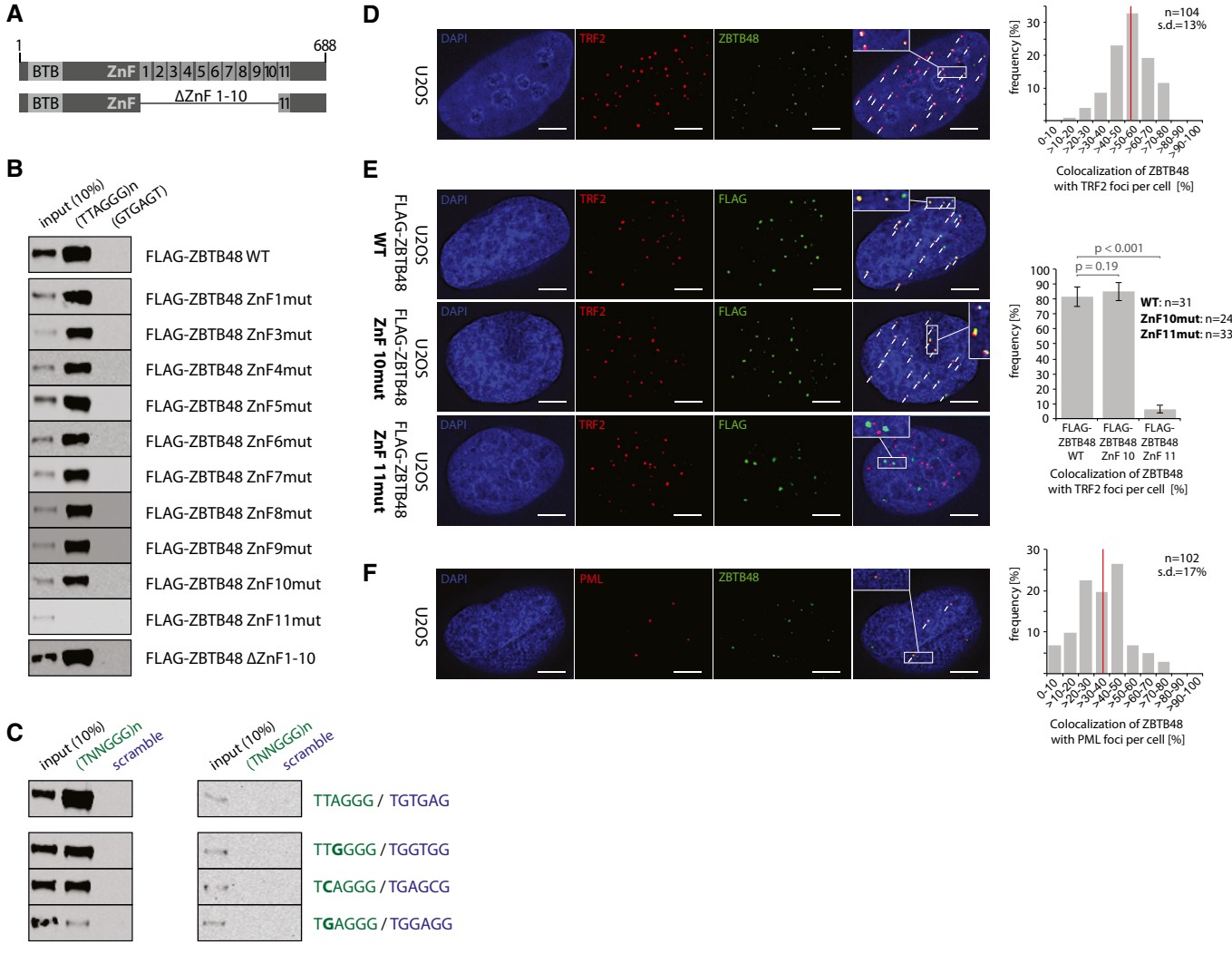

**Figure 1. ZBTB48 binds telomeres via its ZnF11 domain *in vitro* and *in vivo*.**

A   Domain structure of wild-type (WT) ZBTB48 containing a N-terminal BTB domain and 11 zinc fingers (ZnF) at the C-terminus of the 688-amino-acid protein. Note that ZnF2 is degenerate. Below is a schematic of the deletion construct lacking ZnF1-10 used in (B).

B   Sequence-specific DNA pull-downs with either telomeric (TTAGGG) or a control sequence (GTGAGT) for FLAG-ZBTB48 WT, point mutants for the ten functional zinc fingers and a domain deletion construct for ZnF1-10.

C   Sequence-specific DNA pull-downs for FLAG-ZBTB48 WT and ZnF11 point mutant for telomeric and subtelomeric variant repeat sequences (green) and their respective scrambled controls (blue).

D   Co-localization analysis of endogenous ZBTB48 and TRF2 in U2OS cells by immunofluorescence (IF) staining. A representative image illustrating the co-localization between ZBTB48 (green) and TRF2 (red) as a marker for telomeres is shown with DAPI (blue) used as a nuclear counterstain. The quantification of frequency of co-localization events (right) was done after 3D reconstruction of the acquired *z*-stacks (*n* = 104 cells). The average value is indicated by a red bar.

E   IF stainings for exogenous FLAG-ZBTB48 WT and point mutants for ZnF10 and ZnF11 in U2OS cells. The same analysis as in (D) was performed and average co-localization frequencies are shown (*n* = 24–33 cells). Error bars indicate standard deviations, and *P*-values are based on Student's *t*-test.

F   Co-localization analysis between ZBTB48 (green) and PML (red) as a marker for PML bodies analogous to (D) in U2OS cells (*n* = 102 cells).

Data information: (D–F) Co-localization events are indicated by white arrows. Scale bars represent 5 μm.
Source data are available online for this figure.

Similar to a recent report on frequent telomeric localization of overexpressed FLAG-ZBTB48 at telomeres in telomerase-positive cells (HeLa 1.2.11) with long telomeres [14], we also observe co-localization of exogenous FLAG-ZBTB48 WT to about 25% of TRF2 foci in both HeLa 1.3 and HT1080 super-telomerase cells. Again, similar to HeLa Kyoto and U2OS cells, endogenous ZBTB48 localized less frequently to telomeres in HeLa 1.3 and HT1080 super-telomerase, overlapping with about 5% of TRF2 foci in both cell lines (Fig EV1F and G). Overall, these data show that ZBTB48 localizes to telomeres *in vivo* and that the association is more robustly detected in cells with longer telomeres and under over-expression conditions.

## ZBTB48 binds to telomeres in cells with short and long telomeres

To further investigate the *in vivo* binding of ZBTB48 to telomeres and to exclude that ZBTB48 telomere binding in cancer cells with short telomeres is masked in immunofluorescence due to a low abundance at telomeres and/or relatively strong signal of extratelomeric ZBTB48, we performed chromatin immunoprecipitation experiments combined with next-generation sequencing (ChIPseq). To rigorously validate any putative enrichments, we used two independent antibodies against endogenous ZBTB48 and compared enrichment with these antibodies in U2OS and HeLa to any enrichment observed in KO clones of the respective cell lines (Fig EV2A–E). In contrast to the more conventional IgG control, KO cells share the same non-specific enrichment of the antibody and only differ in DNA–protein interactions specifically mediated by the presence of the desired target protein. In agreement with our immunofluorescence data, both ZBTB48 antibodies enriched telomeric DNA from two independent U2OS WT clones in biological and technical duplicates. Confirming the specificity of this telomere binding, in two independent U2OS KO clones both antibodies only showed background levels similar to those observed in IgG control and input samples (Fig 2A). The absolute enrichment of telomeric repeats was lower compared to TRF2, again in agreement with the

partial co-localization observed in U2OS cells. Overall the telomeric enrichment by endogenous ZBTB48 was quantitatively similar compared to that of HOT1 (Figs 2A and EV2F–I, and EV3A), which also localizes to a subset of telomeres [4,9,10]. To rescue U2OS ZBTB48 KO cells, we expressed exogenous FLAG-ZBTB48 WT, which readily binds telomeric DNA. Similar to the co-localization data, overexpression of ZBTB48 enriched more telomeric sequences than the endogenous protein, quantitatively reaching closely the enrichment observed with TRF2 (Fig 2A). As expected, the FLAG-ZBTB48 ZnF11mut failed to associate with telomeric DNA while FLAG-ZBTB48 ZnF10mut behaved quantitatively like the WT construct as observed in co-localization experiments (Figs 1E and 2A). In agreement with this clear enrichment of telomeric DNA, a MEME motif analysis yielded telomeric motifs for ChIPseq reactions with both ZBTB48 antibodies, similar to data obtained with TRF2 and HOT1 (Fig 2B). We next compared enrichment of telomeric DNA in ChIPseq experiments between U2OS and HeLa cells. While the total amount of 50-bp reads containing seven or eight TTAGGG repeats—used here as a stringent cut-off to separate true telomeric reads from interstitial sequences—is higher in ZBTB48 ChIPseq experiments from U2OS, HeLa cells equally show a significant enrichment of telomeric reads with both antibodies compared to their ZBTB48 KO controls (Fig 2C). These data suggest that indeed

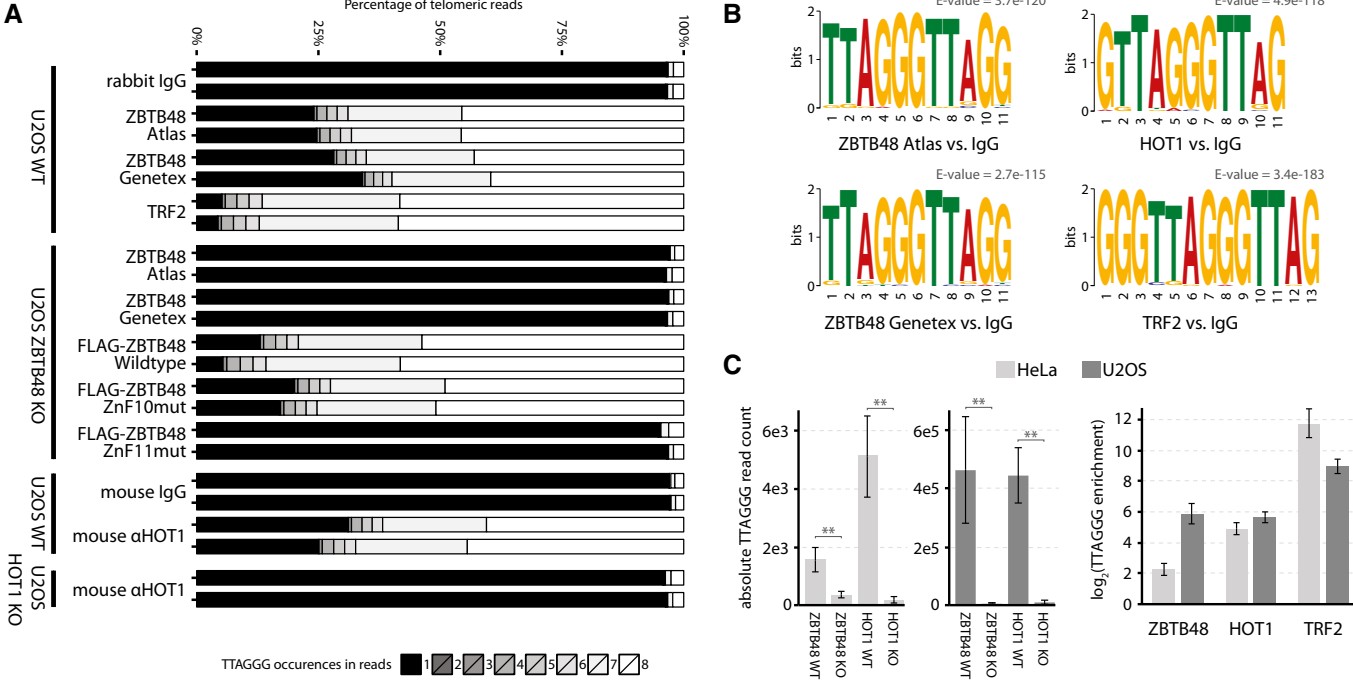

**Figure 2. ZBTB48 associates with short and long telomeres *in vivo*.**

A   TTAGGG content of telomeric reads in ChIPseq samples from U2OS WT, U2OS ZBTB48 KO and U2OS HOT1 KO clones using IgG, TRF2, ZBTB48 Atlas, ZBTB48 GeneTex and HOT1 antibodies as well as FLAG antibody for FLAG-ZBTB48 WT and point mutants for ZnF10 and ZnF11. The percentage contribution to all reads containing 1–8× TTAGGG repeats is shown. All reactions were performed in technical replicates on two independent WT and KO clones each.

B   MEME sequence logo and bit score for the top 500 extratelomeric ChIPseq peaks for ZBTB48 Atlas, ZBTB48 GeneTex, TRF2 and HOT1 antibodies compared to IgG samples in U2OS. For each antibody, the most frequent motif is shown.

C   TTAGGG enrichment in ChIPseq samples with reads with 7× or 8× TTAGGG repeats. Absolute read counts from ChIPseq reactions from each two HeLa and U2OS WT and KO clones with two independent antibodies for ZBTB48 and HOT1 are shown (left). The corresponding fold enrichments are calculated (right) and compared to TRF2 ChIPseq reactions for which fold enrichments are calculated relative to IgG samples. Error bars represent standard deviations ($n = 4$), and $P$-values are based on Student's *t*-test with ** indicating $P < 0.01$.

absolute ZBTB48 levels at HeLa telomeres are low and in immunofluorescence experiments might easily be judged as lack of association. However, the observed enrichment in both HeLa and U2OS cells demonstrates that ZBTB48 is a telomere-binding protein *in vivo* regardless of the mode of telomere maintenance and telomere length.

### ZBTB48 is a negative regulator of telomere length

Based on the association of ZBTB48 with telomeres in HeLa and U2OS cells, we next asked whether ZBTB48 is functionally relevant for telomere homeostasis. By comparing the telomere length of five HeLa WT with five HeLa ZBTB48 KO clones (Fig EV2) by terminal restriction fragment (TRF) length analysis, we observed overall longer telomeres in the absence of ZBTB48. Despite clonal variability, the KO clones have telomeres that are about twice as long as the WT controls (Fig 3A). This effect was not due to any detectable changes in telomerase activity (Fig EV3B). In contrast, five U2OS ZBTB48 KO clones did not show obvious changes in average telomere length based on pulsed-field gel electrophoresis (PFGE)-resolved TRF analysis after extensive culturing (Fig 3B). As U2OS cells have much longer and fairly heterogeneous telomeres, minor differences in telomere length could be masked by an insufficient resolution of their large DNA fragments. To further test functional consequences of ZBTB48 loss in U2OS cells, we quantified C-circles from WT and KO cells as a surrogate for ALT activity. In two out of five clones, we observed a significant reduction of C-circles compared to WT cells, whereas three other clones showed insignificant reductions (Fig 3C). These data demonstrate that at least in telomerase-positive cells ZBTB48 is a negative regulator of telomere length and that even low amounts of ZBTB48 at telomeres are sufficient for suppression of telomere lengthening.

### ZBTB48 acts as a transcriptional activator at a limited set of target genes

To further dissect ZBTB48's cellular functions and to understand a putative crosstalk between telomere length regulation and other downstream effects of ZBTB48 deletion, we further analysed our ChIPseq data sets. Again, using the KO cell lines as a rigorous control and only accepting binding sites that can be validated with two independent ZBTB48 antibodies in two biological and two technical replicates each, we identified ~1,000 unique binding sites in U2OS and ~250 unique binding sites in HeLa cells with a fold enrichment of at least 8, a FDR below 0.05 and at least 100 reads per kilobase (rpk) within the peak region (Fig 4A–F, Datasets EV1 and EV2). These binding sites were highly enriched at gene promoters, with ~70 and ~55% of all peaks residing in the proximal promoter (≤ 1 kb from the transcriptional start site) in U2OS and HeLa, respectively (Fig 4G). In total, 127 binding sites (about 50% of the

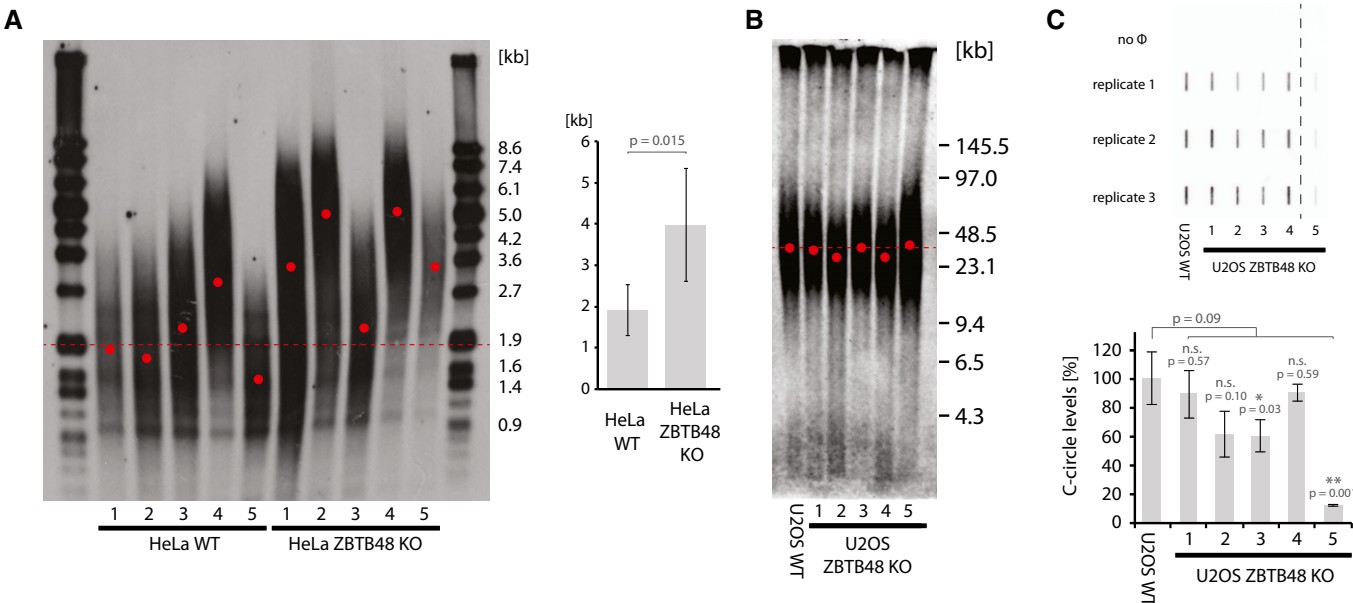

**Figure 3.  ZBTB48 is a negative regulator of telomere length.**

A   Terminal restriction fragment (TRF) analysis for each five independent HeLa WT and HeLa ZBTB48 KO clones at passage 20. The average telomere length was determined from the telomeric distribution (left) and used for average quantification of WT vs. KO clones (right). Average telomere length is indicated for all WT samples (dotted line) and individually for all samples (red dots). Error bars indicate standard deviations, and the *P*-value is based on Student's *t*-test.

B   TRF analysis using pulsed-field gel electrophoresis (PFGE) of parental U2OS WT cells compared to five independent U2OS ZBTB48 KO clones at passage 37. Average telomere length is indicated for all WT samples (dotted line) and individually for all samples (red dots).

C   Quantification of C-circles in U2OS WT cells compared to five independent U2OS ZBTB48 KO clones. C-circle reactions were carried out and spotted in triplicate (top), and average quantifications are displayed (bottom). No Φ indicates negative control reactions without the ΦDNA polymerase. 15 ng DNA was used as input material per reaction. The dashed line indicates cropping of the membrane between KO clones 4 and 5. Error bars indicate standard deviations, and *P*-values are based on Student's *t*-test with * indicating *P* < 0.05 and **"P" < 0.01.

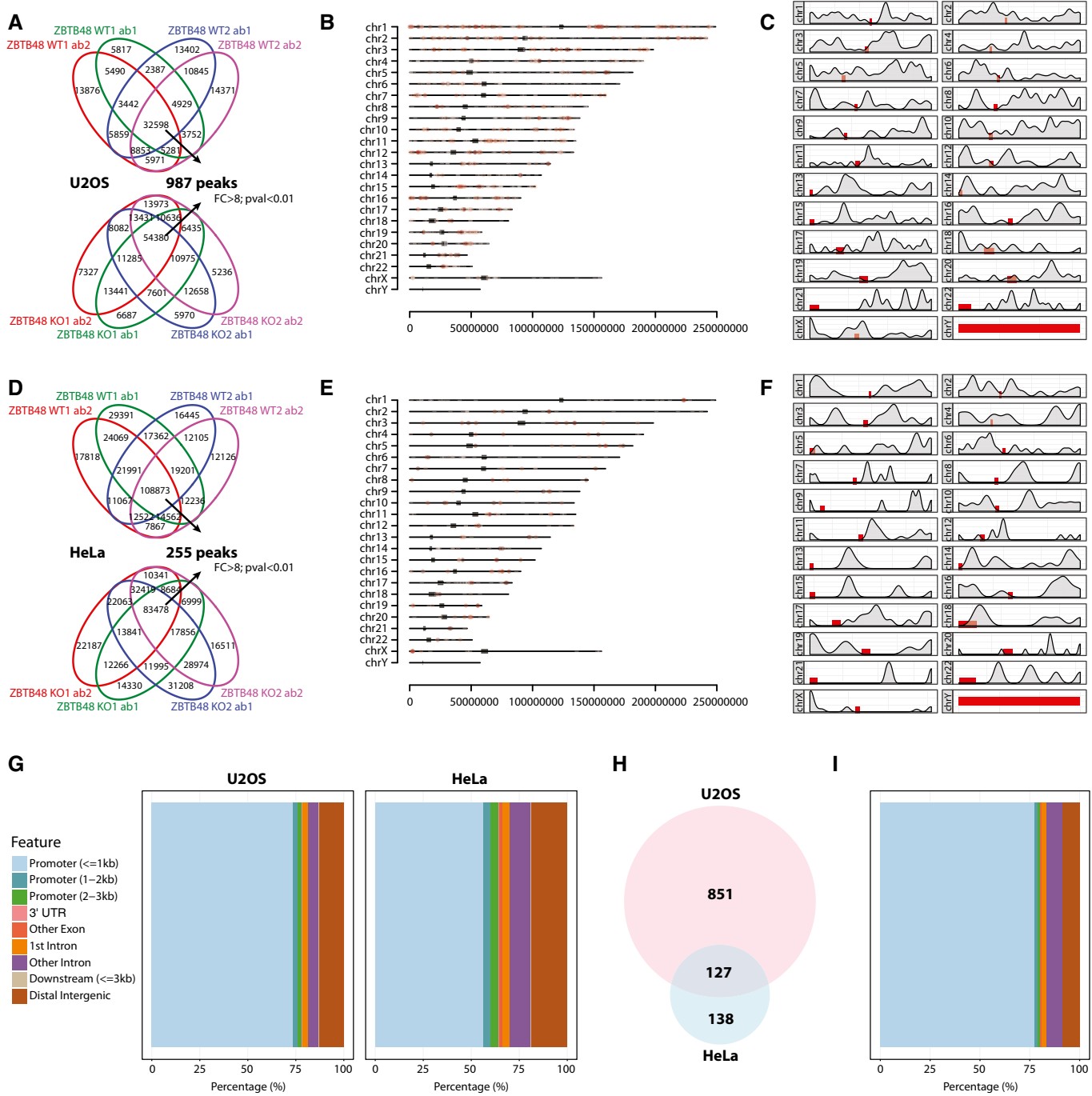

**Figure 4.  ZBTB48 preferentially binds to promoter regions.**

A   Peak calling of ZBTB48 ChIPseq reactions from two U2OS ZBTB48 WT clones (top) and two U2OS ZBTB48 KO clones (bottom). ChIP reactions were performed with two independent ZBT48 antibodies (ab1 and ab2) each in two independent clones (WT1 and WT2 or KO1 and KO2). Only peaks that are enriched by at least eightfold in the WT over the KO clones and have a FDR < 0.01 and are consistently called in all four biological replicates were considered for further analysis.

B   Distribution of ZBTB48 binding sites in U2OS cells along the different chromosomes. Individual peaks are represented by red dots according to their specific chromosomal location, and centromeres are marked as grey blocks.

C   Density distribution of ZBTB48 binding sites along all chromosomes in U2OS cells with centromeres marked as red blocks.

D   Analogous analysis as in (A) for HeLa ZBTB48 WT vs. KO clones.

E   Distribution of ZBTB48 binding sites in HeLa cells along the different chromosomes as in (B).

F   Density distribution of ZBTB48 binding sites along all chromosomes in HeLa cells with centromeres marked as red blocks.

G   Feature distribution of ZBTB48 binding sites in U2OS and HeLa cells. The percentage of ZBTB48 binding sites in promoter, 3′UTR, exonic, intronic, distal intergenic regions as well as in 3′-proximity to genes is shown.

H   Venn diagram of the overlap between ZBTB48 binding sites in U2OS and HeLa cells.

I   Feature distribution of ZBTB48 binding sites as in (G) for overlapping binding sites in U2OS and HeLa cells from (H).

binding sites found in HeLa) overlap between both cell lines (Fig 4H) and these overlapping sites are even more strongly enriched for proximal promoters (Fig 4I).

Its DNA-binding ability and the preferential binding to promoter regions suggest that ZBTB48 might carry out transcriptional regulatory functions. To test whether ZBTB48 indeed impacts on transcriptional levels, we performed next-generation RNA sequencing (RNAseq) analysis comparing each five WT and KO clones as biological replicates for U2OS and HeLa. While the vast majority of transcripts were not significantly changed, 23 and 11 transcripts were significantly downregulated in U2OS and HeLa cells, respectively, based on cut-offs of fold change > 2 and *P*-value < 0.01 (Figs 5A and EV4A, Datasets EV3 and EV4). Four out of the five protein-coding transcripts affected in HeLa were also depleted in U2OS cells. All transcripts in HeLa and all except two in U2OS, the antisense non-coding RNA LA16c-358B7.4 and the lincRNA RP4-758J18.13, were significantly downregulated in the ZBTB48 KO clones, suggesting that ZBTB48 acts as a transcriptional activator on most targets. In comparison, HeLa HOT1 KO cells only show significant downregulation of a single transcript (Fig EV4B, Dataset EV5). To further confirm that ZBTB48 indeed impacts on expression levels, we performed label-free quantitative expression proteomics analysis in HeLa cells quantifying more than 6,000 proteins (Dataset EV6). While for technical reasons proteomes do not yet routinely have the same coverage as RNAseq data, we identified two downregulated proteins, MTFP1 and PXMP2, that actually overlap with the differentially expressed transcripts of the RNAseq experiments (Fig 5B). In agreement with a role as a transcription factor, promoters of 15 out of the 24 protein-coding transcripts affected in ZBTB48 KO cells were bound in ZBTB48 ChIPseq experiments with 11 promoters (MTFP1, PXMP2, SNX15, PPP3CB, VWA5A, CCDC106, TMEM63C, TMEM100 MGAT5, MMP15, RNF150 and TUBA4A) bound in both cell lines (Fig 5C, Datasets EV1 and EV2). Notably, the promoters of MTFP1, PXMP2, SNX15, PP3CB and VWA5A were consistently bound in both U2OS and HeLa cells and their transcript levels were depleted in KO cells of both cell types (Fig 5C and D). Other examples highlight the presence of cell-type-specific differences. For instance, the CCDC106 promoter is clearly under ZBTB48-dependent transcriptional control in U2OS cells, but is not bound by ZBTB48 in HeLa cells. Furthermore, TMEM63C only showed clear expression in U2OS WT cells, although the TMEM63C promoter was occupied by ZBTB48 in both cell lines, suggesting putative priming [24]. In reverse, the MGAT5 promoter was bound both in HeLa and U2OS cells, while we could only detect a clear downregulation of MGAT5 mRNA in HeLa cells (Fig 5C and D). This suggests that some of the additional ZBTB48-bound promoters are either co-regulated by additional transcription factors, require external stimuli or both. Together these data clearly demonstrate that ZBTB48 has transcription factor activity and that it regulates a defined set of target genes.

### ZBTB48 is required for MTFP1 expression

MTFP1 is a nuclear-encoded, mitochondrial protein that has been previously described to regulate mitochondrial fission [25]. As any telomeric phenotypes of ZBTB48 KO cells need to be carefully dissected for direct or indirect effects given, for example, various reports of links between mitochondria and telomeres [7,8], we further investigated the consequences of reduced MTFP1 expression. The MTFP1 promoter was covered by two ZBTB48 ChIPseq peaks in both HeLa and U2OS cells (Fig 5C) and in both cell lines MTFP1 mRNA was significantly reduced in ZBTB48 KO cells (Fig 5A and D) and we could further validate this downregulation using quantitative proteomics (Fig 5B). In agreement with this, we validated the reduced protein levels of MTFP1 by antibody staining for HeLa and U2OS for five WT and KO clones each (Fig 6A). As MTFP1 RNAi knock-down has been described to cause changes in the morphology and distribution of the mitochondrial network [25], we next explored whether ZBTB48 KO cells phenocopy this effect. Using MitoTracker staining and super-resolution Zeiss Airyscan microscopy, we indeed observed defects in the mitochondrial network in both cell lines. While in HeLa WT cells individual mitochondria are easily discernible and the mitochondrial network is spread throughout the entire cytoplasm, mitochondria in HeLa ZBTB48 KO cells are clustered together around the nucleus, recapitulating the previously described loss of MTFP1 phenotype (Figs 6B and EV5A, Movies EV1 and EV2), a phenotype also observable in U2OS ZBTB48 KO cells (Fig EV5B). These changes are not based on differences in the number of mitochondria in WT versus KO cells as judged by quantification of mtDNA levels (Fig 6C). These data show that ZBTB48 is indeed required for MTFP1 expression and that its dual role as a telomere-binding protein and a transcriptional activator provides an interesting link between telomere homeostasis and mitochondrial metabolism. On a larger scale, it highlights the importance of carefully investigating multiple ZBTB48 functions in order to integrate all key aspects of its biology.

## Discussion

In an independent study published in parallel to our work, Li *et al* [14] likewise described ZBTB48 as a novel direct telomere-binding protein. In agreement with our findings, the authors located ZBTB48's TTAGGG-binding ability to the zinc fingers 9–11 using deletion constructs, a notion that we here further refine specifically to ZnF11. In contrast to TRF1, TRF2 and HOT1 that bind preferentially to TTAGGG compared to variant repeat sequences [10,20], ZBTB48 also has affinity for the variant repeats TCAGGG and TTGGGG. This is somewhat reminiscent of the binding behaviour of NR2C/F transcription factors that bind similarly well to TTAGGG and TCAGGG [4,9,21]. In addition to a reported competition for binding sites with TRF2 [14] and other shelterin members, it will be interesting to see how ZBTB48 and the NR2C/F transcription factors relate to each other especially in ALT-positive cancers in which variant repeat sequences such as TCAGGG are amplified and interspersed within the telomeric tract [21].

In their independent study, Li *et al* [14] described robust co-localization of ZBTB48 to telomeres in U2OS cells and in telomerase-positive cells with long telomeres when overexpressing exogenous ZBTB48, but not in telomerase-positive cells with short telomeres. Together with the observation of rapid telomere length reduction in U2OS cells upon ZBTB48 overexpression, longer telomeres in mES ZBTB48$^{-/-}$ cells and the ability of TRF2 overexpression to compete off exogenous ZBTB48, the authors suggest a model in which ZBTB48 binds to long telomeres when TRF2 levels are insufficient for a full coverage. Under these circumstances, ZBTB48 binding

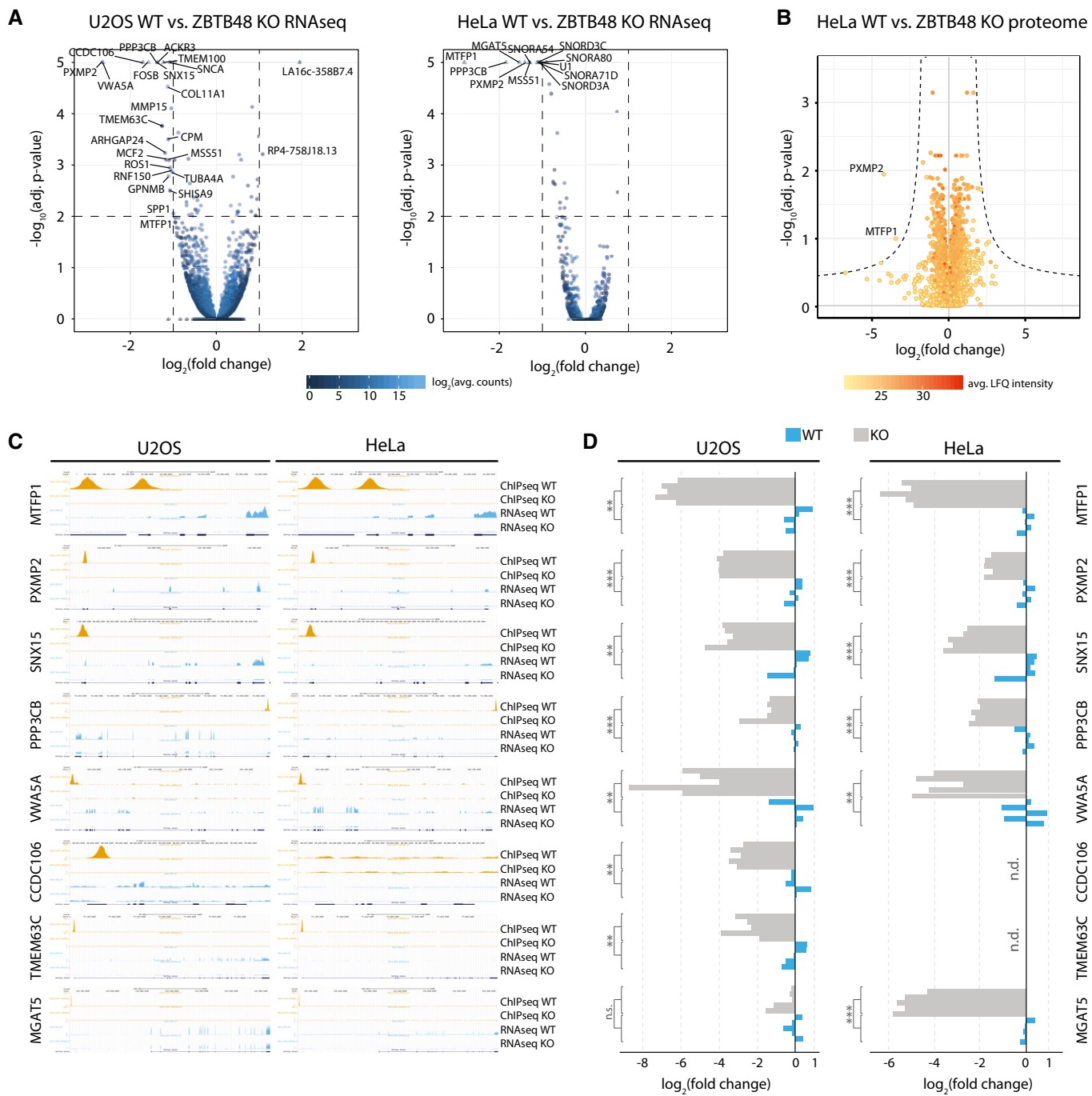

**Figure 5.  ZBTB48 is a transcriptional activator.**

A   Differential expression analysis of the RNA sequencing (RNAseq) gene quantitation, comparing each five WT and ZBTB48 KO clones for U2OS (left) and HeLa (right). Cut-offs for significant differential expression were set to log₂(fold change) > |1| and −log₁₀(adjusted P-value) > 2 (FDR < 0.01). Transcripts with a −log₁₀(adjusted P-value) > 5 are displayed as triangles, and the y-axis is cropped beyond this value.

B   Protein expression analysis comparing two HeLa WT and ZBTB48 KO clones by label-free mass spectrometry. Significantly differential expression was determined based on an empirically determined S₀ and an adjusted P-value <0.05.

C   Representative genome browser tracks for ZBTB48 target genes in U2OS and HeLa cells. ChIPseq tracks for peaks at the promoters of MTFP1, PXMP2, SNX15, PPP3CB, VWA5A, CCDC106, TMEM63C and MGAT5 are shown for WT and KO cells. The corresponding RNAseq tracks for the WT and KO cells are shown.

D   qPCR validations of mRNA expression changes of MTFP1, PXMP2, SNX15, PPP3CB, VWA5A, CCDC106, TMEM63C and MGAT5 comparing each five WT and ZBTB48 KO clones for U2OS (left) and HeLa (right). Each clone was quantified using the average of technical triplicates, and P-values were calculated for the biological quintuplicates (n = 5) based on Student's t-test with ** indicating P < 0.01 and ***P < 0.001. n.d. = not determined; n.s. = not significant.

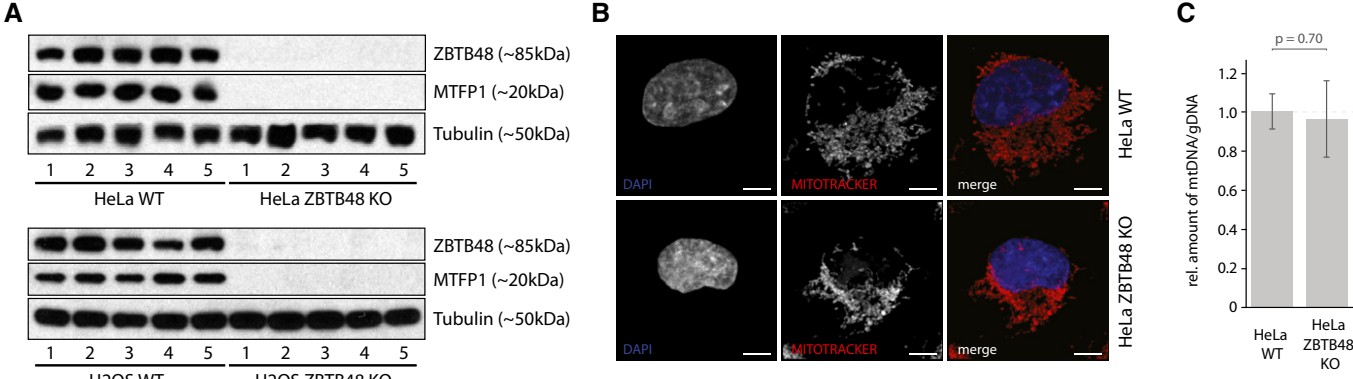

**Figure 6. ZBTB48 is required for MTFP1 expression.**
A  Western blot confirmation of reduced MTFP1 expression in each five HeLa and U2OS ZBTB48 KO clones compared to five WT clones each. Similar to ZBTB48, no detectable MTFP1 protein is found in the KO cells.
B  Super-resolution fluorescence microscopy analysis of the structure and localization of the mitochondrial network in HeLa WT and ZBTB48 KO clones. Mitochondria are marked with the MitoTracker dye (red), and nuclei are counterstained with DAPI (blue). Scale bars represent 5 μm.
C  mtDNA level quantification comparing five HeLa ZBTB48 WT and KO clones. mtDNA levels were quantified based on three mtDNA loci and normalized to two genomic regions. Error bars represent standard deviations ($n$ = 3). The *P*-value is based on Student's *t*-test.

subsequently induces telomere trimming, preventing additional lengthening of telomeres. As an extension to these data, we here show by ChIPseq in HeLa cells that ZBTB48 also binds to short telomeres (Fig 2C). And even though the absolute ZBTB48 abundance at these telomeres seems to be fairly low, the presence of few molecules of ZBTB48 appears to be sufficient to suppress telomere lengthening in these cells, as HeLa ZBTB48 KO clones have longer telomeres than their WT controls (Fig 3A). Notably, the larger abundance of ZBTB48 at U2OS telomeres does not readily translate into further telomere lengthening in the corresponding U2OS ZBTB48 KO clones (Fig 3B). While it is possible that the mechanism underlying the ZBTB48 telomere length regulation also engages telomere trimming in a telomerase-positive context, ZBTB48's function is not limited to control telomere elongation of long telomeres. Thus, this finding extends the previously described role of ZBTB48-dependent telomere length regulation [14] to short telomeres, an important aspect for further studies related to ageing and cancer. Therefore, it seems plausible that cooperatively multiple factors such as the shelterin complex, telomerase, the CST complex and HOT1 define together with ZBTB48 a mean telomere length based on the individual cellular make-up. Interestingly, ZBTB48 has recently been reported as one of five genes that is significantly downregulated in mouse embryonic stem cells between passage 6 and passage 16, coinciding with telomere lengthening during *in vitro* expansion [26]. Furthermore, ZBTB48 localizes to chromosome 1p36, a region that is frequently rearranged (leiomyoma & leukaemia) or deleted (neuroblastoma, melanoma, Merkel cell carcinoma, pheochromocytoma, and carcinomas of colon and breast) in different human cancers and therefore might be a putative tumour suppressor [27–29], but not without dispute [30]. While these rearrangements and deletions require further analysis, they may facilitate an escape from telomere-induced replicate senescence as a putative mechanism to allow sufficient telomere lengthening in some cancers. Interestingly, in contrast to HOT1 KO clones (Fig EV2) most of the ZBTB48 KO clones that we generated have only one detectable ZBTB48 allele,

suggesting loss of heterozygosity (Fig EV2). Likewise, ZBTB48's transcription factor activity could contribute to this as the transcriptional target VWA5A has been described as a tumour suppressor in breast cancer, nasopharyngeal carcinoma and melanoma [31–33] and MGAT5 and CCDC106 regulate PTEN and TP53, respectively [34,35]. In this context, it will be important to dissect direct telomeric phenotypes of ZBTB48 from indirect effects, which both might involve additional factors.

Our unbiased multi-layer omics analysis already revealed an additional, non-telomeric function of ZBTB48 as a transcriptional activator. ZBTB48 has been previously implicated in transcriptional activation of the ARF gene in HEK293 cells [36]. While we did not detect any changed ARF mRNA levels nor ZBTB48 binding to the ARF promoter neither in HeLa nor in U2OS cells, this may be due to differences in cell-type-specific target genes. This is in agreement with the fact that beyond overlapping examples such as MTFP1, PXMP2, SNX15, PPP3CB and VWA5A, we also observe different ZBTB48 target genes in HeLa and U2OS, such as CCDC106, and overall we observe more ZBTB48 binding sites in U2OS cells. This transcription factor activity links ZBTB48 to various biological processes. For instance, PXMP2 for which downregulation in ZBTB48 KO cells could be validated both on mRNA and protein levels is a peroxisomal membrane protein that is important for normal mammary gland development [37,38]. Furthermore, the virtually complete dependence of MTFP1 expression on ZBTB48 strengthens the link between telomeres and mitochondria. It has previously been shown that telomere dysfunction compromises mitochondrial metabolism [7] and that in reverse conplastic mouse strains divergent by multiple mutations/variants in their mtDNA also differed in telomere length as well as their rate of telomere shortening [8]. Additionally, a telomerase mutant defective in nuclear-cytoplasmic shuttling induces mitochondrial dysfunction even in a context of long telomeres [39]. These examples are further extended by reports on RAP1 affecting transcription of key mitochondrial regulators such as PPARα and PGC1α [40] and key

telomeric proteins, such as TIN2 and TERT, localizing to mitochondria [41,42]. In this context, ZBTB48 represents an additional link between telomere binding and the mitochondrial network that strengthens the previously described interplay [7]. It will be interesting to see how these diverse phenotypes affect each other and whether, for example, increased ZBTB48 tethering to telomeres compromises MTFP1 expression. This duality between telomere binding and transcription factor activity is reminiscent of TRF2 occupancy at interstitial binding sites and its transcription factor activity [43–45], potentially representing an important and frequent mechanism of how telomeres crosstalk with other cellular processes. Furthermore, it will be interesting to see whether binding to interstitial sites and the subsequent transcriptional regulation involve the multitude of remaining zinc fingers that are not necessary for telomere binding.

In sum, we here describe ZBTB48 as a novel telomere-binding protein that recognizes telomeric and subtelomeric variant repeats. While on the one hand ZBTB48 limits telomere elongation, it also acts as a transcriptional activator on a defined set of target genes including the mitochondrial fission regulator MTFP1. Thus, our findings intrinsically connect both telomere and mitochondrial homeostasis.

# Materials and Methods

### Cell culture

HeLa Kyoto, HeLa 1.3, U2OS and HT1080 super-telomerase cells were cultivated in 4.5 g/l glucose, 4 mM glutamine, 1 mM sodium pyruvate containing Dulbecco's modified Eagle's medium (DMEM) supplemented with 10% foetal bovine serum (FBS; Sigma), 100 U/ml penicillin and 100 μg/ml streptomycin (Gibco) in a humidified incubator at 37°C and 5% $CO_2$.

### Nuclear protein extraction

For preparation of nuclear extracts, cells were harvested and incubated in hypotonic buffer (10 mM Hepes, pH 7.9, 1.5 mM $MgCl_2$, 10 mM KCl) on ice for 10 min. Cells were transferred to a Dounce homogenizer in hypotonic buffer supplemented with 0.1% Igepal CA-630 (Sigma) and 0.5 mM DTT and subjected to 40 strokes. Nuclei were washed once in 1× PBS and extracted in hypertonic buffer [420 mM NaCl, 20 mM Hepes, pH 7.9, 20% glycerol, 2 mM $MgCl_2$, 0.2 mM EDTA, 0.1% Igepal CA-630 (Sigma), 0.5 mM DTT, complete protease inhibitor (Roche)] for 2 h at 4°C on a rotating wheel.

### Telomere pull-down

25 μg of forward and reverse sequence oligonucleotides (Table 1) were diluted in annealing buffer (20 mM Tris–HCl, pH 7.5, 10 mM $MgCl_2$, 100 mM KCl), denatured at 95°C, and annealed by cooling. Annealed double-strand oligonucleotides were incubated with 100 units T4 kinase (Life Technologies) for 2 h at 37°C followed by incubation with 20 units T4 ligase overnight. Concatenated DNA strands were purified using phenol–chloroform extraction. Following biotinylation with desthiobiotin-dATP (Jena Bioscience) and 60 units DNA polymerase (Thermo Scientific), the biotinylated probes

were purified using microspin G-50 columns (GE Healthcare). Telomeric or control DNA was immobilized on 375 μg paramagnetic streptavidin beads (Dynabeads MyOne C1, Thermo Scientific) on a rotation wheel for 30 min at room temperature. Subsequently, baits were incubated with 40 μg nuclear extract in PBB buffer (150 mM NaCl, 50 mM Tris–HCl pH 7.5, 5 mM $MgCl_2$, 0.5% Igepal CA-630 (Sigma) and 1 mM DTT) while rotating for 2 h at 4°C. 20 μg sheared salmon sperm DNA (Ambion) was added as a competitor for DNA binding. After three washes with PBB buffer, bound proteins were eluted in 2× Laemmli buffer (Sigma-Aldrich), boiled for 5 min at 95°C and separated on a 4–12% NuPAGE Novex Bis–Tris precast gel (Thermo Scientific).

### Deletion/mutant variant construction

The ZBTB48 clone was obtained from the ORFeome collection (IOH40784, Invitrogen). The sequence was LR recombined into a Gateway-compatible pcDNA3.1 vector with N-terminal FLAG-tag (gift from Christian Brandts). Deletion variants were constructed by PCR amplification of the pcDNA3.1-FLAG vector using primers with site-specific overhangs (Metabion) following the QuikChange II site-directed mutagenesis kit protocol (Stratagene) [46]. Point mutants were generated similarly using the QuikChange protocol using primers shown in Table 2. The constructs were sequence-verified using an Applied Biosystems 3730 Genetic Analyzer (Applied Biosystems) according to the manufacturer's instructions.

### Plasmid transfection

For plasmid transfections in HeLa, cells were seeded in 10-cm dishes the evening before transfection ($1.5 × 10^6$ cells corresponding to 55 $cm^2$). 5 μg DNA was transfected with 20 μl Lipofectamine 2000 (Invitrogen) according to the manufacturer's instructions.

For TALEN transfection in HeLa, cells were seeded in six-well dishes ($1.5 × 10^5$ cells corresponding to 9 $cm^2$). 2 μg of the ZBTB48 TALEN pair (H12250) or HOT1 TALEN pair (H75088; both from TALEN Library Resource, Seoul National University [47]) were transfected with 5 μl Lipofectamine 2000 according to the manufacturer's instructions. For plasmid transfection in U2OS, the Amaxa Cell Line Nucleofector Kit V (Lonza) was used according to the manufacturer's instructions. 2 or 6 μg DNA were used for $1 × 10^6$ cells or $1 × 10^7$ cells, respectively. After electroporation (X001), cells were transferred to pre-warmed medium.

### Western blot

Western blot samples were boiled in Laemmli buffer (Sigma-Aldrich) at 95°C for 5 min and run on a 4–12% Bis–Tris gel (NuPAGE, Thermo Scientific). Blotting to a nitrocellulose membrane (Protram; Schleicher&Schuell) was performed using a semi-dry transfer system for 90 min at 50 mA. The membrane was blocked in PBS containing 0.1% Tween-20 (PBST) and 5% (w/v) non-fat milk for 1 h at RT and incubated with primary antibody overnight at 4°C. The following primary antibodies were used: rabbit anti-ZBTB48 (HPA030417, Atlas antibodies, Sigma-Aldrich, 1:1,000), mouse anti-FLAG M2 (F1804, Sigma-Aldrich, 1:5,000), rabbit anti-MTFP1 (SAB4301167, Sigma-Aldrich, 1:1,000), mouse anti-DM1alpha tubulin (MPI-CBG Antibody Facility, 1:50,000

**Table 1.  Oligonucleotides used for pull-down experiments.**

| No | Sequence motif | Primer sequence (5′→3′) |
|---|---|---|
| 1a | TTAGGG for | TTAGGGTTAGGGTTAGGGTTAGGGTTAGGGTTAGGGTTAGGGTTAGGGTTAGGGTTAGGG |
| 1b | TTAGGG rev | AACCCTAACCCTAACCCTAACCCTAACCCTAACCCTAACCCTAACCCTAACCCT |
| 2a | GTGAGT for | GTGAGTGTGAGTGTGAGTGTGAGTGTGAGTGTGAGTGTGAGTGTGAGTGTGAGTGTGAGT |
| 2b | GTGAGT rev | ACACTCACACTCACACTCACACTCACACTCACACTCACACTCACACTCACACTC |
| 3a | TTGGGG for | TTGGGGTTGGGGTTGGGGTTGGGGTTGGGGTTGGGGTTGGGGTTGGGGTTGGGGTTGGGG |
| 3b | TTGGGG rev | AACCCCAACCCCAACCCCAACCCCAACCCCAACCCCAACCCCAACCCCAACCCC |
| 4a | TGGTGG for | GTGGTGGTGGTGGTGGTGGTGGTGGTGGTGGTGGTGGTGGTGGTGGTGGTGGTGGTGGTG |
| 4b | TGGTGG rev | ACCACCACCACCACCACCACCACCACCACCACCACCACCACCACCACCACCACC |
| 5a | TCAGGG for | GTCAGGGTCAGGGTCAGGGTCAGGGTCAGGGTCAGGGTCAGGGTCAGGGTCAGGGTCAGG |
| 5b | TCAGGG rev | ACCCTGACCCTGACCCTGACCCTGACCCTGACCCTGACCCTGACCCTGACCCTG |
| 6a | TGAGCG for | GTGAGCGTGAGCGTGAGCGTGAGCGTGAGCGTGAGCGTGAGCGTGAGCGTGAGCGTGAGC |
| 6b | TGAGCG rev | ACGCTCACGCTCACGCTCACGCTCACGCTCACGCTCACGCTCACGCTCACGCTC |
| 7a | TGAGGG for | GTGAGGGTGAGGGTGAGGGTGAGGGTGAGGGTGAGGGTGAGGGTGAGGGTGAGGGTGAGG |
| 7b | TGAGGG rev | ACCCTCACCCTCACCCTCACCCTCACCCTCACCCTCACCCTCACCCTCACCCTC |
| 8a | TGGAGG for | GTGGAGGTGGAGGTGGAGGTGGAGGTGGAGGTGGAGGTGGAGGTGGAGGTGGAGGTGGAG |
| 8b | TGGAGG rev | ACCTCCACCTCCACCTCCACCTCCACCTCCACCTCCACCTCCACCTCCACCTCC |
| 9a | TTAGGC for | TTAGGCTTAGGCTTAGGCTTAGGCTTAGGCTTAGGCTTAGGCTTAGGCTTAGGCTTAGGC |
| 9b | TTAGGC rev | AAGCCTAAGCCTAAGCCTAAGCCTAAGCCTAAGCCTAAGCCTAAGCCTAAGCCT |
| 10a | GTGACT for | GTGACTGTGACTGTGACTGTGACTGTGACTGTGACTGTGACTGTGACTGTGACTGTGACT |
| 10b | GTGACT rev | ACAGTCACAGTCACAGTCACAGTCACAGTCACAGTCACAGTCACAGTCACAGTC |
| 11a | TTAGG for | TTAGGTTAGGTTAGGTTAGGTTAGGTTAGGTTAGGTTAGGTTAGGTTAGGTTAGGTTAGG |
| 11b | TTAGG rev | AACCTAACCTAACCTAACCTAACCTAACCTAACCTAACCTAACCTAACCTAACCT |
| 12a | GTAGT for | GTAGTGTAGTGTAGTGTAGTGTAGTGTAGTGTAGTGTAGTGTAGTGTAGTGTAGTGTAGT |
| 12b | GTAGT rev | ACACTACACTACACTACACTACACTACACTACACTACACTACACTACACTACACT |
| 13a | TCAGG for | GTCAGGTCAGGTCAGGTCAGGTCAGGTCAGGTCAGGTCAGGTCAGGTCAGGTCAG |
| 13b | TCAGG rev | ACCTGACCTGACCTGACCTGACCTGACCTGACCTGACCTGACCTGACCTG |
| 14a | GTAGC for | GTAGCGTAGCGTAGCGTAGCGTAGCGTAGCGTAGCGTAGCGTAGCGTAGCGTAGC |
| 14b | GTAGC rev | ACGCTACGCTACGCTACGCTACGCTACGCTACGCTACGCTACGCTACGCTACGCT |

dilution) and goat anti-GAPDH (AP16240PU-N, Acris Antibodies, 1:5,000). After three washes in 5% milk PBST for 10 min each at RT, the membrane was incubated for 20–30 min at RT with secondary antibodies (donkey anti-rabbit and donkey anti-mouse IRDye 800CW as well as donkey anti-goat IRDye 680RD LI-COR Odyssey, 1:15,000; goat anti-rabbit antibody conjugated to horseradish peroxidase, Bio-Rad, 1:4,000) in 5% milk PBST, followed by two washes in 5% milk PBST, two washes in PBST and one in PBS for 10 min each at RT. The membrane was revealed either with the LI-COR Odyssey imaging system or with Pierce ECL Western blotting substrate (Thermo Scientific). Spectra Multicolor Broad Range Protein Ladder (Fermentas), PageRuler Plus Prestained Protein Ladder (Thermo Scientific) and MagicMark XP Western Protein Standard (Invitrogen) were used as standards.

**T7 Endonuclease 1 (T7E1) assay**

After TALEN transfection, cells were split and partly used for genomic DNA extraction using the QIAamp DNA Blood Mini Kit

(Qiagen) following the manufacturer's instructions. A PCR amplifying the spacer locus was carried out using the primers shown in Table 3. For one reaction, 10 µl 5× HF buffer, 1.5 µl 50 mM MgCl$_2$, 1 µl 25 mM dNTP mix, 2 µl 10 mM primer each, 0.5 µl Phusion High-Fidelity DNA Polymerase (NEB) and 100 ng genomic DNA were used, filled up to 50 µl with HPLC-grade H$_2$O. PCR was performed under following conditions: initial hotstart denaturation at 98°C for 5 min, followed by 35 cycles with 98°C for 30 s, 60°C for 30 s, 72°C for 30 s and finished by a final extension step at 72°C for 5 min in a PCR cycler (Peqlab). PCR products were analysed by gel electrophoresis and purified using QIAquick PCR purification kit (Qiagen).

100 ng each of purified WT and TALEN-treated DNA were denatured at 95°C for 5 min in 1× NEB 2 buffer and hybridized following the temperature profile 95 to 85°C at 2°C/s and 85 to 25°C at 0.1°C/s. For digestion of heteroduplexes, 2 µl of 10× NEB 2 buffer, 1 µl T7 Endonuclease I (NEB) and 7 µl HPLC-grade H$_2$O were added, and samples were incubated at 37°C for 20 min and analysed on a 2% agarose gel.

**Table 2.  Oligonucleotides used for QuikChange mutagenesis.**

| No | Sequence motif | Primer sequence (5′→3′) |
|---|---|---|
| 1a | ZnF1 H>A mutation for | GCAAATATTATCTAAAAGTCGCCAACAGGAAACATACTGGGGAGAAACCC |
| 1b | ZnF1 H>A mutation rev | TCCCCAGTATGTTTCCTGTTGGCGACTTTTAGATAATATTTGCTGAGGAA |
| 2a | ZnF3 H>A mutation for | GAAGGATGGAGCTGCGGGTGGCCATGGTGTCTCACACAGGGGAGATGCCC |
| 2b | ZnF3 H>A mutation rev | TCCCCTGTGTGAGACACCATGGCCACCCGCAGCTCCATCCTTCGGCGGAA |
| 3a | ZnF4 H>A mutation for | AGAAGAAGGACTTGCAGAGCGCCATGATCAAACTTCATGGAGCCCCCAAG |
| 3b | ZnF4 H>A mutation rev | GCTCCATGAAGTTTGATCATGGCGCTCTGCAAGTCCTTCTTCTGCATGAA |
| 4a | ZnF5 H>A mutation for | CTCGGACAGAGCTGCAGCTGGCTGAAGCTTTCAAGCACCGTGGTGAGAAG |
| 4b | ZnF5 H>A mutation rev | CCACGGTGCTTGAAAGCTTCAGCCAGCTGCAGCTCTGTCCGAGACAGGAA |
| 5a | ZnF6 H>A mutation for | GCCGGAATGGCCTGCAGATGGCCATCAAGGCCAAGCACAGGAATGAGAGG |
| 5b | ZnF6 H>A mutation rev | TTCCTGTGCTTGGCCTTGATGGCCATCTGCAGGCCATTCCGGCTCGAGGC |
| 6a | ZnF7 H>A mutation for | AAAAGGCCAATCTCAACATGGCCCTGCGCACACACACGGGTGAGAAGCCC |
| 6b | ZnF7 H>A mutation rev | TCACCCGTGTGTGTGCGCAGGGCCATGTTGAGATTGGCCTTTTGGGTGAA |
| 7a | ZnF8 H>A mutation for | CCCAAGCCAGCCTGGACAAGGCCAACCGCACCCACACCGGGGAAAGGCCC |
| 7b | ZnF8 H>A mutation rev | TCCCCGGTGTGGGTGCGGTTGGCCTTGTCCAGGCTGGCTTGGGTTCGGAA |
| 8a | ZnF9 H>A mutation for | AGAAGGGGCCCCTCCTGAGGGCCGTGGCCAGCCGCCATCAGGAGGGCCGG |
| 8b | ZnF9 H>A mutation rev | TCCTGATGGCGGCTGGCCACGGCCCTCAGGAGGGGCCCCTTCTCAGTGAA |
| 9a | ZnF10 H>A mutation for | CCGTGGAGCAACTGCGTGTGGCCCGTCAGACGGCACAAGGGGGTGAGGAAG |
| 9b | ZnF10 H>A mutation rev | ACCCCCTTGTGCCGTCTGACGGCCACACGCAGTTGCTCCACGGCTTTGAA |
| 10a | ZnF11 H>A mutation for | GACAGGCCCACCTGCGGAGGGCCATGGAGATCCACGACCGGGTAGAGAAC |
| 10b | ZnF11 H>A mutation rev | ACCCGGTCGTGGATCTCCATGGCCCTCCGCAGGTGGGCCTGTCGGGTAAA |
| 11a | ΔZnF 1-3 mutation for | GGGCATCTCCCCTGTCGGCACCGCTGTACCTTTTCTGTTC |
| 11b | ΔZnF 1-3 mutation rev | GGTACAGCGGTGCCGACAGGGGAGATGCCCTACAAGTGTT |
| 12a | ΔZnF 4-6 mutation for | TGGCCTCTCATTCCTGGCATCTCCCCTGTGTGAGACACCA |
| 12b | ΔZnF 4-6 mutation rev | CACAGGGGAGATGCCAGGAATGAGAGGCCACACGTATGTG |
| 13a | ΔZnF 7-9 mutation for | AGGAATGAGAGGCCACAGGAGGGCCGGCCCCACTTCTGCC |
| 13b | ΔZnF 7-9 mutation rev | GGGCCGGCCCTCCTGTGGCCTCTCATTCCTGTGCTTGGCC |
| 14a | ΔZnF 7-11 mutation for | AGGAATGAGAGGCCAGACCGGGTAGAGAACTACAACCCGC |
| 14b | ΔZnF 7-11 mutation rev | GTTCTCTACCCGGTCTGGCCTCTCATTCCTGTGCTTGGCC |
| 15a | ΔZnF 1-9 mutation for | GGTACAGCGGTGCCGCAGGAGGGCCGGCCCCACTTCTGCC |
| 15b | ΔZnF 1-9 mutation rev | GGGCCGGCCCTCCTGCGGCACCGCTGTACCTTTTCTGTTC |
| 16a | ΔZnF 1-10 mutation for | GGTACAGCGGTGCCGAAGGGGGTGAGGAAGTTTGAGTGCA |
| 16b | ΔZnF 1-10 mutation for | CTTCCTCACCCCCTTCGGCACCGCTGTACCTTTTCTGTTC |

**Table 3.  Oligonucleotides used for TALEN validation.**

| No | Sequence motif | Primer sequence (5′→3′) |
|---|---|---|
| 1a | ZBTB48 TALEN validation primer for | CTGGCTTCGCTGAGATCTTT |
| 1b | ZBTB48 TALEN validation primer rev | CCTGGGCACAGTACCTCATT |
| 2a | HOT1 TALEN validation primer for | GGGCTTATTGCACTCCATGT |
| 2b | HOT1 TALEN validation primer rev | GCGTCTGTGTGGAAGCTGTA |

### Genotyping of TALEN clones

For genotyping of screened single cell KO clones, PCR product was cloned in pCR2.1-TOPO TA using the same amplification protocol as for the T7E1 assay above and transformed into bacteria (TOPO TA

cloning kit, Thermo Scientific) according to the manufacturer's instructions using blue/white screening. At least nine white colonies were picked for each cell line clone, and DNA was purified prior to sequencing.

### Immunofluorescence staining and fluorescence microscopy

Immunofluorescence stainings were performed on glass coverslips (0.17 mm, assorted glass, Thermo Scientific) or in 96-well plates. Cells were fixed with 10% formalin for 10 min at RT, followed by two washes with 1× PBS + 30 mM glycine. Cells were permeabilized in 1× PBS + 0.5% Triton X-100 for 5 min at 4°C, again followed by two washes with 1× PBS + 30 mM glycine. After blocking for 15 min in blocking solution (0.2% fish skin gelatine (Sigma-Aldrich) in

1× PBS) at RT, primary antibodies were diluted in blocking solution and incubated for 1 h at RT. The following antibodies were used: rabbit anti-ZBTB48 (HPA030417, Atlas antibodies, Sigma-Aldrich, 1:500), mouse anti-FLAG M2 (F1804, Sigma-Aldrich, 1:500), mouse anti-TRF2 (NB100-56506, Novus Biologicals, 1:250), mouse anti-PML (sc-966, Santa Cruz, 1:500) and mouse anti-HOT1 (MPI-CBG Antibody Facility, 1:1,000). Cells were washed three times for 3 min each in blocking solution. Next, samples were incubated with fluorescent-labelled donkey anti-rabbit-IgG and donkey anti-mouse-IgG (both from Invitrogen) with either Alexa488 or Alexa594 as secondary antibodies (1:500 in blocking solution) for 30 min at RT. After three additional washes in blocking solution, cells were either treated with 1 μg/ml DAPI and kept in 1× PBS (96-well plate) or mounted using DAPI Prolong Gold Antifade Reagent (Thermo Scientific).

For mitochondrial staining, cells were grown on coverslips and incubated with 50 nM of MitoTracker dye (Thermo Scientific) for 30 min in a humidified incubator at 37°C and 5% $CO_2$. Cells were washed once with 1× PBS and fixed with 10% formalin for 10 min followed by two washes with 1× PBS. For permeabilization, cells were incubated 1× PBS + 0.5% Triton X-100 for 5 min at RT and washed twice with 1× PBS. Cells were mounted using Vectashield with DAPI (Vector laboratories).

Imaging for co-localization experiments was conducted on a DeltaVision Core Microscope (Applied Precision, Olympus IX71) with a 100×/1.4 UPlanSApo oil-immersion objective and *z*-stacks (0.2-μm optical sections) were acquired. Deconvolution was performed with softWoRx (Applied Precision). *Z*-stacks were reconstructed in 3D using Imaris (Bitplane), and co-localization events were determined for signals above the background using the co-localization function. For the analysis of mitochondrial distribution, imaging was conducted on a Zeiss LSM 800 with Airyscan (Carl Zeiss) with a 63×/1.4 PlanApo oil-immersion objective and *z*-stacks (0.2-μm optical sections) were acquired or on a Zeiss Imager M2 with a 40×/0.75 Plan Neofluor objective.

## Next-generation chromatin immunoprecipitation sequencing (ChIPseq)

Subconfluent HeLa and U2OS cells were washed twice with cold 1× PBS and double cross-linked first with 2 mM DSG (Pierce, predissolved in DMSO) in 1× PBS for 45 min at RT, intermitted by a wash with cold 1× PBS and secondly with 1% formaldehyde in DMEM for 20 min at RT. The formaldehyde was quenched with glycine (0.125 M final concentration) for 5 min at RT, and cells were scraped with a plastic scraper and washed once in 1× PBS.

For lysis, cells were resuspended in buffer 1 (50 mM Tris–HCl pH 8.0, 250 mM sucrose, 140 mM NaCl, 1 mM EDTA pH 8.0, 10% glycerol, 0.5% Igepal CA-630 (Sigma-Aldrich), 0.25% Triton X-100, 0.25% Tween-20) for 15 min, pelleted and subsequently resuspended in buffer 2 (10 mM Tris–HCl pH 8.0, 200 mM NaCl, 1 mM EDTA pH 8.0, 0.5 mM EGTA pH 8.0) for 10 min both at 4°C. Sonication was performed in 1.5 ml sonication buffer (1% SDS, 10 mM EDTA pH 8.0, 50 mM Tris–HCl pH 8.0) with a Branson Digital Sonifier S-450D for 7.5 min (30% amplitude, 10 s ON, 20 s OFF).

Per IP, 125 μl of paramagnetic protein G-coated beads (Dynabeads, Invitrogen) were washed three times in modified PBB [180 mM NaCl, 50 mM Tris–HCl pH 8.0, 5 mM $MgCl_2$, 0.25% Igepal CA-630 (Sigma-Aldrich), 1 mM DTT, complete protease inhibitor without EDTA (PI, Roche)] with a supplement of 10 μg/μl BSA (NEB) and for the second wash of 100 μg sheared salmon sperm DNA (Ambion). 50 μl beads were used to preclear 50 μg lysate diluted in modified PBB (180 mM NaCl) for 1 h at 4°C. After removal of beads, 5 μg of antibody was added and incubated on a rotating wheel overnight. The following antibodies were used: rabbit anti-ZBTB48 (HPA030417, Atlas antibodies, Sigma-Aldrich), rabbit anti-ZBTB48 (GTX118671, GeneTex), mouse anti-FLAG M2 (F1804, Sigma-Aldrich), rabbit anti-TRF2 (NB110-57130, Novus Biologicals), mouse anti-HOT1 (MPI-CBG Antibody Facility) and rabbit anti-HOT1 (MPI-CBG Antibody Facility) as well as rabbit and mouse IgG. The 75 μl of the remaining beads were added and incubated for 2 h. Beads were washed seven times with PBB (150 mM NaCl, without PI) and once with TE buffer (pH 8.0). Samples were eluted twice with 100 μl elution buffer (1% SDS, 0.1 M $NaHCO_3$) and de-cross-linked with 8 μl 5 M NaCl at 65°C. To remove RNAs and proteins, samples were digested with 100 μg RNase A and 40 μg proteinase K.

For DNA purification, paramagnetic beads (Agencourt AMPure XP, Beckman Coulter) were used according to the manufacturer's instructions, and DNA concentrations were measured with Pico-Green (Thermo Scientific). Library preparation was performed with the NuGEN Ovation Ultralow Library System V2 1-96. Libraries were prepared with a starting amount of 124–6,869 pg of ChIP DNA, and 10–145 ng of input DNA was amplified in 13–18 PCR cycles. Libraries were profiled on a 2100 Bioanalyzer (Agilent) and quantified using the Qubit dsDNA HS Assay Kit on a Qubit 2.0 Fluorometer (Thermo Scientific). Size selection was performed with Lab ChIP XT with DNA 750 chips (PerkinElmer) for the 300- to 500-bp fragment range according to the manufacturer's instructions. Libraries were pooled in equimolar ratio and sequenced on a HiSeq (Illumina), 50-bp single-end read in high-output mode plus 7 cycles for the index read.

## Next-generation RNA sequencing

Exponentially growing HeLa and U2OS WT, ZBTB48 KO and HOT1 KO cells were harvested and RNA was extracted using the RNeasy Mini Kit (Qiagen) with on-column DNaseI digestion according to the manufacturer's instructions. RNA quality was assessed by a 2100 Bioanalyzer (Agilent), and only samples with a RIN score of 10 were subjected to RNAseq library preparation. Library preparation was performed with the TruSeq stranded mRNA LT sample prep kit (Illumina). rRNA was removed from 1 μg of total RNA using the Ribo-Zero rRNA removal kit (Epicentre). rRNA-depleted RNA was purified using Agencourt RNA Clean XP beads (Beckman Coulter) and eluted in 8.5 μl of elution buffer. Sample preparation was continued with the elute, prime, fragment step (but skipping the polyA selection steps) of the TruSeq stranded mRNA protocol until the end of library preparation, using 12 PCR cycles for library amplification. Libraries were quality-controlled on a 2100 Bioanalyzer with a DNA 1000 chip (Agilent) and quantified using the Qubit dsDNA HS Assay Kit on a Qubit 2.0 Fluorometer (Thermo Scientific). Samples were pooled in equimolar ratio and sequenced on a HiSeq 2500 (Illumina), 50-bp paired-end read in high-output mode plus 7 cycles for the index read.

## Telomere restriction fragment length analysis

For HeLa cells, genomic DNA was extracted using the QIAamp DNA Blood Mini Kit (Qiagen) according to the manufacturer's instructions. Average telomere length was measured by Southern blot analysis of terminal restriction fragments using the TeloTAGGG telomere length assay kit (Sigma-Aldrich) following the manufacturer's instructions with slight modifications [48]. 2–5 µg of DNA was digested using 20 U of HinfI and RsaI each at 37°C for 2 h, and the digested DNA was resolved on an 0.8% agarose gel at 120 V for 4 h in 1× TAE buffer. For depurination, the gel was incubated in 0.25 M HCl for 30 min, followed by two rinses with distilled water. The gel was washed twice with denaturation solution (0.5 M NaOH, 1.5 M NaCl) for 20 min each. Subsequently, the gel was rinsed with distilled water twice followed by two washes with neutralizing solution (0.5 M Tris–HCl, 3 M NaCl, pH 7.5) twice for 20 min each. The DNA was then transferred to a nylon membrane (Hybond, $N^+$, Amersham, UK) overnight by capillary osmosis using 20× SSC (3 M NaCl, 0.3 M sodium citrate tribasic dehydrate, pH 7). Fixing of DNA onto the membrane was carried out by UV-cross-linking at 120 mJ using a Stratalinker® UV Crosslinker (Stratagene), followed by two rinses with 2× SSC. Following prehybridization for 1 h at 42°C using pre-warmed DIG Easy Hyb Granules, 1 µl DIG-labelled telomere probe/5 ml was added for hybridization and incubated for 3 h. The membrane was washed twice with 100 ml stringent buffer 1 (2× SSC, 0.1% SDS), 5 min each at RT, followed by two washes with 100 ml pre-warmed stringent buffer 2 (0.2× SSC, 0.1% SDS), 20 min each at 50°C. After rinsing with 1× wash buffer for 5 min, the membrane was blocked with 100 ml 1× blocking solution provided in the kit for 1 h and then incubated with anti-DIG-AP antibody (1:10,000) diluted in blocking solution. Subsequently, the membrane was washed twice with 100 ml 1× washing buffer, 15 min each at RT, and then incubated in 100 ml 1× detection solution for 5 min at RT. The TRF smear was detected using the digoxigenin luminescent detection (CDP-Star) system and X-ray films. Average telomere length was calculated by comparison to the 1 kb plus DNA ladder provided in the kit.

For U2OS, cells were washed twice in 1× PBS and resuspended in HIRT buffer without detergent (10 mM Tris pH 7.6, 100 mM NaCl, 10 mM EDTA) supplemented with 200 µg RNase. For lysis, HIRT buffer with SDS (final SDS concentration 0.5%) was added and incubated for 60 min at 37°C before digest with proteinase K at 55°C overnight. DNA purification was carried out with phenol–chloroform extraction in 5Prime Phase Lock tubes (VWR). 7.5 µg of U2OS DNA was digested with 40 U HinfI and RsaI each supplemented with 0.3 µg RNase in 1× NEB2 at 37°C overnight. 5 µg of digested DNA was loaded onto a 1% SeaKem Gold agarose gel (Lonza) in 0.5× TBE placed in a CHEF electrophoresis chamber. The low-range PFG marker (NEB) was used as size standard. Electrophoresis was conducted with initial switch time 0.2 s and final switch time 13 s at 6 V/cm for 15 s. After EtBr visualization, the gel was submerged in alkaline transfer solution (0.4 N NaOH, 0.6 M NaCl) for 30 min. Capillary transfer with alkaline transfer solution on a Hybond $N^+$ membrane (Amersham) was done overnight. The membrane was recovered and subsequently incubated in 0.4 N NaOH for 15 min and 2× SSC for 5 min twice before UV-cross-linking.

The membrane was prehybridized in hybridization buffer [6× SSC, 0.1% SDS, 1% milk (w/v)] for 1 h at 42°C. A telomeric oligonucleotide (TAA(CCCTAA)$_4$) was labelled with DIG using the 3'-End Labeling Kit (Roche) according to the manufacturer's instructions and boiled at 99°C for 10 min in 6× SSC. Hybridization with the DIG labelled oligonucleotide was performed in hybridization buffer at 42°C overnight. The membrane was washed twice for 5 min in 2× SSC/0.1% SDS and once for 2 min in 0.2× SSC/0.1% SDS each at 42°C, before transfer to 2× SSC at room temperature. The membrane was incubated with maleic acid buffer (0.1 M maleic acid, 0.15 M NaCl, pH adjusted to 7.5 with NaOH) supplemented with 0.3% Tween-20 for 5 min, blocking buffer [maleic acid buffer supplemented with 1% milk (w/v)] for 30 min and antibody solution (blocking buffer with anti-DIG-AP antibody, Roche, 1:20,000) for 30 min before washing twice with maleic acid buffer supplemented with 0.3% Tween-20 for 15 min each and addition of detection buffer (0.1 M Tris, 0.1 M NaCl, pH adjusted to 9.5 with HCl) for 5 min. CDP-Star solution (Roche) was added before scan.

## C-circle assay

DNA extraction and digestion were performed as described for TRF. Subsequently, 500 ng DNA was amplified with 7.5 units ΦDNA polymerase (Fermentas) in 1× Φ29 buffer supplemented with 2 mM dATP, dGTP, dTTP (Invitrogen) and 0.1 mg/ml BSA for 12 h at 30°C. After heat inactivation, products were denatured in 0.4 M NaOH, 0.01 M EDTA and slot-blotted on a Biodyne B nylon membrane (Pall). NaOH was neutralized with 2× SSC, and DNA was UV-cross-linked onto the membrane. The hybridization was conducted as described for TRF.

## qPCR assays for validation of ZBTB48 target genes and mtDNA quantification

For expression analysis, RNA was extracted using the RNEasy Plus Mini Kit (Qiagen) with on-column DNaseI digestion following the manufacturer's instructions. Genomic DNA was isolated using the QIAamp DNA Blood Mini Kit (Qiagen) following the manufacturer's instructions. 1 µg of RNA was used for cDNA synthesis using the SuperScript IV first-strand synthesis kit with oligo-dT following the manufacturer's protocol. 1 µg of RNA was used for cDNA synthesis using the SuperScript IV first-strand synthesis kit with oligo-dT following the manufacturer's protocol. PCR was carried out using 1x QunatiNova SYBR green PCR mix (Qiagen) and 500 nM each of the forward primer and reverse primer (Table 4) with the following cycling conditions on a CFX96 Real-Time PCR machine (Bio-Rad): 95°C for 2 min, followed by 40 cycles with 95°C for 5 s, 60°C for 30 s followed by a melting curve. TBP was used as a housekeeping gene for normalization.

The relative mtDNA copy number was quantified by normalizing the copy number measurement of three mitochondrial genes, MT-CYB (Cytochrome B), MT-COX1 (Cytochrome C oxidase 1) and MT-ATP6 (ATP synthase 6), against a nuclear gene, ATP5B (ATP synthase subunit beta), and a nuclear gene desert region of chromosome 10, using the primers shown in Table 5. PCR was carried out using 20 ng of DNA, 1× QunatiNova SYBR green PCR mix (Qiagen) and 200 nM each of the forward primer and reverse primer with the following cycling conditions on a CFX96 Real-Time PCR machine

**Table 4.  Oligonucleotides used for expression validation of ZBTB48 target genes.**

| No | Sequence motif | Primer sequence (5′→3′) |
|----|----------------|-------------------------|
| 1a | MTFP1 for | GCTGTTGACCATCCCCATCA |
| 1b | MTFP1 rev | ACTGTTGGGTAGAGCTTGCG |
| 2a | PXMP2 for | GTGGGCCTCTGAGATATGCC |
| 2b | PXMP2 rev | AGAAGAAGTGACTCAGCGGC |
| 3a | SNX15 for | CACGTCTTGCTTCAGGGAGT |
| 3b | SNX15 rev | GGAGTTGAGACAGGTGCAGG |
| 4a | PPP3CB for | GAGTGTTAGCTGGAGGACGG |
| 4b | PPP3CB rev | AGCCTCAATAGCCTCAACTGT |
| 5a | VWA5A for | CGTGGTTTTGGAGATGGGGA |
| 5b | VWA5A rev | CGAGCGGTCCATGAGAAAGA |
| 6a | CCDC106 for | CAGAAGGGAGGTGCTAGTCG |
| 6b | CCDC106 rev | GGAAGGTGCCCAGGATCTTC |
| 7a | TMEM63C for | ATGGAACGCAGAGACAAGGG |
| 7b | TMEM63C rev | CGTCGTCCCCACACTTGTTA |
| 8a | MGAT5 for | CTCCTATGGACGGCTACCCT |
| 8b | MGAT5 rev | AGGGATCTGAACGCCACATG |
| 9a | TBP for | TTCGGAGAGTTCTGGGATTG |
| 9b | TBP rev | CTCATGATTACCGCAGCAAA |

**Table 5.  Oligonucleotides used for mtDNA quantification.**

| No | Sequence motif | Primer sequence (5′→3′) |
|----|----------------|-------------------------|
| 1a | Mitochondria-encoded CytB for | TATCCGCCATCCCATACATT |
| 1b | Mitochondria-encoded CytB rev | GTGTGAGGGTGGGACTGTCT |
| 2a | Mitochondria-encoded CoxI for | CCTGACTGGCATTGTATTAGCA |
| 2b | Mitochondria-encoded coxi rev | AGTGGAAGTGGGCTACAACG |
| 3a | Mitochondria-encoded ATP6 for | GCCCTAGCCCACTTCTTACC |
| 3b | Mitochondria-encoded ATP6 rev | CAGGGCTATTGGTTGAATGAG |
| 4a | Nuclear-encoded ATP5B for | AAGCTGTGGCAAAAGCTGAT |
| 4b | Nuclear-encoded ATP5B rev | TAGGGGCAAGGAGAGAGACA |
| 5a | Nuclear gene desert region chr10 for | GGCTAATCCTCTATGGGAGTCTGTC |
| 5b | Nuclear gene desert region chr10 rev | CCAGGTGCTCAAGGTCAACATC |

(Bio-Rad): 95°C for 2 min, followed by 40 cycles with 95°C for 5 s, 60°C for 10 s.

**MS data acquisition**

For in-gel digestion, samples were reduced in 10 mM DTT (Sigma) for 1 h at 56°C followed by alkylation with 55 mM iodoacetamide (Sigma) for 45 min in the dark. Tryptic digest was performed in 50 mM ammonium bicarbonate buffer with 1 µg trypsin (Promega) at 37°C overnight. Peptides were desalted on self-made C18 Stage Tips and analysed with a Q Exactive Plus mass spectrometer (Thermo) coupled to an EASY-nLC 1000 system (Thermo) nanoflow liquid chromatography system. Peptides were separated on a C18 reversed-phase capillary (25 cm long, 75 µm inner diameter, packed in-house with ReproSil-Pur C18-AQ 1.9 µm resin (Dr. Maisch) directly mounted on the electrospray ion source. We used a 4.5 h gradient from 2 to 40% acetonitrile in 0.5% formic acid at a flow of 225 nl/min. The Q Exactive Plus was operated in positive ion mode with a Top10 MS/MS spectra acquisition method per MS full scan. MS scans were obtained with 70,000 resolution at a maximum injection time of 20 ms and MS/MS scans at 17,500 resolution with maximum IT of 120 ms and using HCD fragmentation. Peptides with unassigned or charge state 1 were excluded, peptide match was preferred, and dynamic exclusion was set to 40 s.

**MS data analysis**

The raw files were processed with MaxQuant [49] version 1.5.2.8 against the UNIPROT annotated human protein database (81,194 entries). Carbamidomethylation on cysteines was set as fixed modification while methionine oxidation and protein N-acetylation were considered as variable modifications. Peptide and protein FDR were enforced at 0.01 with match between runs option activated. LFQ quantitation [50] was made on unique peptides for protein groups with at least 2 ratio counts.

**Bioinformatic analysis**

The protein group table was further processed with an in-house R script excluding contaminants (according to the contaminant list of MaxQuant), reverse hits and proteins only identified by a modified peptide. The biological and technical replicates were combined for statistical analysis and subjected to a two-tailed Welch's *t*-test to calculate *P*-values for each protein group. For scoring of differentially expressed proteins, an empirically determined $S_0$ and an adjusted *P*-value < 0.05 (indicated by a dashed line in the volcano plot) were chosen.

For the RNAseq experiments with HeLa cells, we obtained between 44 and 71 million of 51-bp paired-end reads in biological quintuplicates. For U2OS cells, we obtained between 30 and 33 million of 68-bp single-end reads, also in biological quintuplicates. Reads were mapped to the human reference genome version GRCh38 together with the accompanying gene model v84 from Ensembl, using STAR version 2.5.1b [51] and allowing up to 2 mismatches. Only uniquely mapped reads were used to quantify expression of genes, using featureCounts v1.4.6-p2 [52] with default parameters and the same gene model used for mapping. Differential expression analysis was carried out using R (http://www.R-project. org/) and DESeq2 [53], which were used to normalize and model the data with generalized linear models (GLM) with the negative binomial link function.

For the identification of interstitial binding sites by ChIPseq experiments, we sequenced two biological replicates using two different antibodies, in duplicated technical conditions. The raw reads from the technical replicated experiments were pooled together, and

we treated the different antibodies as different biological conditions. Thus, we worked with libraries from quadruplicated biological conditions for the knockout (KO) and wild-type (WT) conditions. We obtained between 38 and 63 million of 51-bp single reads after pooling technical replicates together. Reads were mapped to the human reference genome version GRCh38 using Bowtie [54] and allowing up to two mismatches which do not exceed a base quality of 70, keeping only alignments from the best stratum. From the mapped data, we produced bigwig tracks normalized to the number of mapped reads, and we proceeded to call peaks with MACS version 2.1.0.20150420 [55] using the default *q*-value cut-off (0.05) and mfold (5, 52) parameters. In order to remove unspecific peaks, we performed a quantitative differential binding analysis with Diffbind (http://bioconductor.org/packages/release/bioc/vignettes/DiffBind/inst/doc/DiffBind.pdf) [56] between the KO and WT conditions, considering only peaks which were significantly different between the two conditions (FDR < 0.01 and fold change > 8) and called consistently in all four WT replicates. The resulting peak set was annotated with the closest TSS using the gene model v84 from Ensembl.

Identification of sequencing reads containing telomeric repeats was performed on unfiltered sequences, using a simple in-house developed Python script employing regular expressions matching. For motif discovery, we used MEME-ChIP [57] with the sequences from the peak regions and default parameters for the top 500 extratelomeric ChIPseq peaks.

### Data availability

The ChIPseq, RNAseq and mass spectrometry proteomics data have been deposited to the Gene Expression Omnibus with the data set identifier GSE96778. The mass spectrometry proteomics data have been deposited to the ProteomeXchange Consortium via PRIDE [58] with the data set identifier PXD006074.

**Expanded View** for this article is available online.

### Acknowledgements
TALENs were kindly provided by the Seoul National University, South Korea, HT1080 super-telomerase cells by Joachim Lingner and HeLa 1.3 cells by Titia de Lange. We thank Lígia Pina for support with generating TALEN KO cells. Support by the IMB Genomics Core Facility is gratefully acknowledged. Research in the Kappei laboratory was supported by the National Research Foundation Singapore and the Singapore Ministry of Education under its Research Centres of Excellence initiative and by the RNA Biology Center at CSI Singapore, NUS, from funding by the Singapore Ministry of Education's Tier 3 grants, grant number MOE2014-T3-1-006. Work in the Butter group was supported by the DFG (Bu2996/1) and the Rhineland Palatinate Forschungsschwerpunkt GeneRED (Gene Regulation in Evolution and Development). The Buchholz laboratory was supported by the DFG grants SFB655 (TPB05), BU1400/3-1 and BU1400/5-1, and the TUD Excellence initiative by the German Federal and State Governments (Institutional Strategy, measure "support the best"). A.J. was supported by a Mildred Scheel doctoral program scholarship from the German Cancer Aid. The Kumar lab was supported by grants from the National Medical Research Council of Singapore and the National Research Foundation Singapore and the Singapore Ministry of Education under its Research Centers of Excellence initiative.

### Author contributions
DK and FBut initiated the research; AJ, GR, AB, ID and DK planned and performed experiments; MP-R and SS performed large-scale data analysis; CTH prepared RNAseq and ChIPseq libraries; APK and JAL-V contributed to the planning of the experiments; DK, FBuc and FBut supervised the research and contributed to the planning of the experiments; AJ and DK wrote the manuscript with input from all authors.

### Conflict of interest
The authors declare that they have no conflict of interest.

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
