## [Review Process File · EMBO Reports]

Manuscript EMBO-2017-44095

ZBTB48 is both a vertebrate telomere-binding protein and a transcriptional activator

Arne Jahn, Grishma Rane, Maciej Paszkowski-Rogacz, Sergi Sayols, Alina Bluhm, Chung-Ting Han, Irena Draškovič, J. Arturo Londoño-Vallejo, Alan Prem Kumar, Frank Buchholz, Falk Butter, and Dennis Kappei

Corresponding authors: Frank Buchholz (TU Dresden), Falk Butter (Institute of Molecular Biology, Mainz), Dennis Kappei (National University of Singapore)

Review timeline:	Submission date:	16 February 2017
	Editorial Decision:	02 March 2017
	Revision received:	20 March 2017
	Editorial Decision:	22 March 2017
	Revision received:	31 March 2017
	Accepted:	18 April 2017

Editor: Esther Schnapp

Transaction Report:

1st Editorial Decision

02 March 2017

Thank you for the transfer of your research manuscript to EMBO reports. We have now received the full set of referee reports as well as cross-comments that are pasted below.

As you will see, all referees acknowledge that the findings are interesting. However, referee 2 raises a number of technical concerns. Given the related paper published in Science more than one month ago, we can only offer to publish your manuscript if we can proceed with it quickly. Referee 2 points out that telomere length regulation by TZAP in HeLa cells needs to be confirmed by additional data. While referees 1 and 3 agree in their cross-comments that it should be examined whether TZAP associates more with longer telomeres in HeLa clones with long telomeres, they also think that the major point 1 by referee 2 does not need to be addressed. The other comments from referee 2 should be addressed as suggested by referee 3 in the cross-comments. All concerns by referee 1 should be addressed too.

Please let me know whether you think that these revisions can be performed within 3-4 weeks. If we can accept your paper before the 22nd of March, we will be able to include it in our May issue. Otherwise we would publish it online as soon as possible.

We would thus like to invite you to revise your manuscript with the understanding that the referee concerns must be addressed, also in a complete point-by-point response. Acceptance of the manuscript will depend on a positive outcome of a second, short round of review. It is EMBO reports policy to allow a single round of revision only and acceptance or rejection of the manuscript

will therefore depend on the completeness of your responses included in the next, final version of the manuscript.

REFEREE REPORTS

Referee #1:

Here the authors describe a similar effect of ZBTB48 at telomeres to that shown in a recent Science publication. They also reveal a novel finding of this protein as a transcriptional regulator. The manuscript is acceptable for publication but I would suggest the authors make a few small changes relating to the tone of the manuscript that in parts are over written - rather than being accurate and succinct. Here are some examples:

1. In the introduction, authors state that large-scale screens had little overlap. The nature of these screens were quite different. In the context of other screens (siRNAs, CRISPR etc) the degree of overlap was actually comparably high. So the author's statement is misleading.
2. Authors should reference Li et al., 2017 directly after citing the name TZAP as it was this manuscript that initially described ZBTB48's role at telomeres and re-named it as TZAP.
3. Pg. 6. "Compared to a recent report....." When citing a difference with an existing observation the authors should state explicitly here whether the 2 studies utilized the same cell lines or not. It's not the same experiment otherwise.
4. Figure 1 D-F. An enlarged merged insert would be helpful for visual clarity.
5. Relating to Figure 2D-E. I would caution the authors from stating that there is no effect on U2OS cells based on this TRF analysis. U2OS telomeres should be resolved and quantified by PFGE as in Dilley et al., Nature 2016. Also, they should label the lanes 1, 2, 3, 4 etc.
6. Figure 1C. Blots on right look terrible and highly processed. Should be replaced with better quality blots.

In addition, the authors declare and discuss in relative detail a minor conflict with the Li et al. 2016 regarding telomere length. This is based on observations with different HeLa clones. I would suggest that rather than make this an issue of contention, why don't they just go and test this in HeLa with long telomeres (HeLa 1.2.11 for example) and determine whether there is a greater association of TZAP with longer telomeres. The authors should do this. It's such a trivial experiment - and would be better for the field to resolve this here.

Referee #2:

In this interesting paper, the authors characterized the protein ZBTB48 (TZAP) that they previously identified as a telomeric protein. TZAP was recently described in a report in Science. Nonetheless, this clearly independent discovery of ZBTB48 as a telomeric protein is of great interest to the field. The authors first characterized the telomere binding property of the protein and demonstrated by mutation analysis that the last zinc finger domain (Zn11) is important for its *in vitro* binding to telomeric and subtelomeric sequences and its *in vivo* recruitment to telomere. They also suggested the potential function of TZAP in regulating the length of bulk telomeres although this reviewer is not entirely swayed by the data which is derived from clonal cell lines. These data are mostly consistent with the data published in the prior paper. The authors went on to characterize the non-telomeric, transcriptional function of TZAP. They found that the protein binds to various loci in the genome and most of the binding sites correspond to the promoter of genes. By RNAseq and MS, they show that various RNA and protein levels are changed after TZAP KO. Among them, they identified MTFP1, a mitochondrial protein, and showed that upon loss of TZAP, MTFP1 is depleted and some defects in mitochondria are reported. This latter part of the work merits more investigation. With suitable revisions, this manuscript would be excellent for publication in EMBO Report.

Major comments:

Telomere length regulation/Figure 2D & 2E:

To determine the role of TZAP in regulating telomere length, the authors do KOs of TZAP and look at telomere length in KO and wt clones. HeLa cells have a lot of clonal variation in telomere length, as the authors acknowledge, and as is shown in examination of their wt clones. Although the authors do a lot of work (analyzing many clones), this reviewer is very concerned that the results may be misleading. One out of five wt clones have long telomeres, whereas three out of five KO clones have long telomeres. Are we sure that if ten clones were examined for each, the numbers may not be much closer (say 4/10 and 5/10?). This is the inherent problem with analyzing telomere length in a clonal setting. To provide definitive evidence for the very important point that TZAP regulates telomere length in HeLa, the authors need to do a bulk population study over a number of PDs. This could be done with an shRNA to TZAP, or a bulk CRISPR KO (which now is possible). Without this data or other corroborating data (e.g. OE of TZAP showing telomere shortening), this important point is insufficiently supported by experimental evidence.

Transcriptional profiling / Figure 3:

This reviewer is very confused by the fact that the ChIP with TZAP on the KO U2OS or HeLa cells reveals more or less the same number of peaks as the TZAP wt cells. How is this possible? What are the controls? Are the peaks normalized to the input sequence? To a non-specific antibody ChIP? (minor comment: the peaks in panel C seem unusually wide; the chromosomal loci of the peaks are not particularly of interest and could be given in a supplemental figure)

MTFP1 regulation and telomere biology:

There are several problems with these experiments that need to be addressed before publication. First, the western blots on MTFP1 show no protein at all in absence of TZAP. But the mass spec shows that there is a 8fold reduction, which would lead to a visible MTFP1 band. Similarly, in the U2OS setting, RNAseq shows that the MTFP1 mRNA level is only 2fold down, yet the western blot shows no protein.

The analysis of mitochondrial status using mitotracker is underdeveloped since only two cells are shown (one wt and one KO). These two cells appear to be in different stages of the cell cycle, which could lead to different mitotracker patterns. Please provide a DIC images to show the whole cell and perform the analysis on at least 20 cells to make sure that cell cycle stages are equally represented in the KO and wt. Also, an shRNA control for the MTFP1 knockdown phenotype (in the wt HeLa clone) and a rescue of the KO cells with wt TZAP are needed to validate the observed results. Without these controls, the results could potentially be spurious.

In addition, In Tondera et al., observed sother phenotypes associated with MTFP1 KD such as cytochrome c release, increased apoptotic rate, reduced proliferation etc. These parameters (at least some) should also be checked.

Even with the additional controls and data, the link between telomere biology and mitochondrial biology is not well established and the text should be modified to include the possibility that TZAP simply has two unrelated roles, one in mitochondrial fusion and one in telomere biology.

Minor comments:

Title and throughout: please refer to ZBTB48 and ZBTB48/TZAP so that the naive reader does not get confused.

Similarly for HOT1, which has a second name used in the Dejardin paper that is cited, the second name should be given throughout. A search for HOT1 would not yield anything other than the papers by this group.

Figure 1B: The nomenclature of the mutants throughout the paper is confusing. FLAG-ZBTB49 ZnF11 sounds like a mutant with ZnF11 domain only instead of a complete protein with a single aa change in ZnF11. Please call it ZnF11mut or something similar.

Figure 1D: According to the Li et al. Science paper, after OE of TZAP in U2OS cells, there should be big blobs of TRF2 and TZAP, why is it not seen here?

Figure 1D: Please indicate which antibody they used to visualize TZAP (anti-FLAG or TZAP). I suppose they used anti-FLAG. A western blot is needed to show expression levels of the different

forms of TZAP.

Figure S1: Please provide quantification for the experiments with HeLa and HT1080 ST cells and a western blot to compare the endogenous expression level of TZAP in the three cell lines tested.

Figure 2A & Figure S3: Why are the ChIPs quantified as a percentage of telomeric reads per total reads? This is not appropriate for ChIP. The read numbers should be normalized to the total telomeric reads in the input, not total reads in the ChIP. This is a standard approach for telomeric ChIPs.

Figure 2C right panel: In order to compare the "enrichment" of TTAGGG in pulldown of two different cell lines, one has to normalize against the length of telomere. Otherwise, cells with longer telomeres will be likely to give higher enrichment because they have more telomeric DNA to be pulled down. If my understanding is correct, the 16x difference between HeLa and U2OS cells in terms of TZAP binding is actually not significant considering that telomeres in U2OS they use is about 16x longer (Figure 2D & E). Hence, maybe the occupancy (or density) of the protein on telomere is the same between HeLa and U2OS. The reason that TZAP foci at telomere are not observed in HeLa cells may be that the telomeres are too short (fewer protein at telomere but with the same density).

Referee #3:

Jahn et al document additional TZAP functions not detailed by the authors' competitors in their recent Science paper. Importantly, these authors report that endogenous TZAP is present on short telomeres (as opposed to completely missing in the Science paper) and unappreciated roles as a transcription factor in mitochondria. Although its role in mitochondria is not well documented, this observation nevertheless could be important, given the previous observation by the DePinho lab linking critical telomere shortening to mitochondrial defects.

Cross-comments from referee 1:

I understand the concerns of reviewer #2 which are well made. However, examining bulk telomere length over time also comes with problems (selection mainly). So, in my opinion, the authors have adhered to the emerging and correct methodology when dealing with CRISPR KO/shRNA variation (See Dilley et al., Nature 2016). In light of the fact that the great majority of the data presented in relation to telomere regulation are in line with the previous study, I don't think this is a major issue. It's okay to have some differences that can be reasoned for. For this reviewers minor issues, these are mostly semantic/basic and can easily be fixed.

For my own point, TZAP was reported as being the first de facto "counter" of telomere length which is pretty important. So I think if the authors can show that they come to their conclusions by doing the same experiments as those shown in the Li et al. paper then I think everyone would be well served.

Cross-comments from referee 3:

Referee #1

Here the authors describe a similar effect of ZBTB48 at telomeres to that shown in a recent Science publication. They also reveal a novel finding of this protein as a transcriptional regulator. The manuscript is acceptable for publication but I would suggest the authors make a few small changes relating to the tone of the manuscript that in parts are over written - rather than being accurate and succinct. Here are some examples:

1. In the introduction, authors state that large-scale screens had little overlap. The nature of these screens were quite different. In the context of other screens (siRNAs, CRISPR etc) the degree of overlap was actually comparably high. So the author's statement is misleading.

2. Authors should reference Li et al., 2017 directly after citing the name TZAP as it was this manuscript that initially described ZBTB48's role at telomeres and re-named it as TZAP.
3. Pg. 6. "Compared to a recent report....." When citing a difference with an existing observation the authors should state explicitly here whether the 2 studies utilized the same cell lines or not. It's not the same experiment otherwise.
4. Figure 1 D-F. An enlarged merged insert would be helpful for visual clarity.
5. Relating to Figure 2D-E. I would caution the authors from stating that there is no effect on U2OS cells based on this TRF analysis. U2OS telomeres should be resolved and quantified by PFGE as in Dilley et al., Nature 2016. Also, they should label the lanes 1, 2, 3, 4 etc.
6. Figure 1C. Blots on right look terrible and highly processed. Should be replaced with better quality blots.

In addition, the authors declare and discuss in relative detail a minor conflict with the Li et al. 2016 regarding telomere length. This is based on observations with different HeLa clones. I would suggest that rather than make this an issue of contention, why don't they just go and test this in HeLa with long telomeres (HeLa 1.2.11 for example) and determine whether there is a greater association of TZAP with longer telomeres. The authors should do this. It's such a trivial experiment - and would be better for the field to resolve this here.

I agree with Reviewer 1 that this exp should be done-it also address rev 2's point 1.

Referee #2

In this interesting paper, the authors characterized the protein ZBTB48 (TZAP) that they previously identified as a telomeric protein. TZAP was recently described in a report in Science. Nonetheless, this clearly independent discovery of ZBTB48 as a telomeric protein is of great interest to the field. The authors first characterized the telomere binding property of the protein and demonstrated by mutation analysis that the last zinc finger domain (Zn11) is important for its in vitro binding to telomeric and subtelomeric sequences and its in vivo recruitment to telomere. They also suggested the potential function of TZAP in regulating the length of bulk telomeres although this reviewer is not entirely swayed by the data which is derived from clonal cell lines. These data are mostly consistent with the data published in the prior paper. The authors went on to characterize the non-telomeric, transcriptional function of TZAP. They found that the protein binds to various loci in the genome and most of the binding sites correspond to the promoter of genes. By RNAseq and MS, they show that various RNA and protein levels are changed after TZAP KO. Among them, they identified MTFP1, a mitochondrial protein, and showed that upon loss of TZAP, MTFP1 is depleted and some defects in mitochondria are reported. This latter part of the work merits more investigation. With suitable revisions, this manuscript would be excellent for publication in EMBO Report.

Major comments:

Telomere length regulation/Figure 2D & 2E:

To determine the role of TZAP in regulating telomere length, the authors do KOs of TZAP and look at telomere length in KO and wt clones. HeLa cells have a lot of clonal variation in telomere length, as the authors acknowledge, and as is shown in examination of their wt clones. Although the authors do a lot of work (analyzing many clones), this reviewer is very concerned that the results may be misleading. One out of five wt clones have long telomeres, whereas three out of five KO clones have long telomeres. Are we sure that if ten clones were examined for each, the numbers may not be much closer (say 4/10 and 5/10)? This is the inherent problem with analyzing telomere length in a clonal setting. To provide definitive evidence for the very important point that TZAP regulates telomere length in HeLa, the authors need to do a bulk population study over a number of PDs. This could be done with an shRNA to TZAP, or a bulk CRISPR KO (which now is possible). Without this data or other corroborating data (e.g. OE of TZAP showing telomere shortening), this important point is insufficiently supported by experimental evidence.

See above-no need to do bulk telomere studies over time if they could use IF to document that TZAP localizes to long telomeres in HeLa1.2.11.

Transcriptional profiling/ Figure 3:

This reviewer is very confused by the fact that the ChIP with TZAP on the KO U2OS or HeLa cells reveals more or less the same number of peaks as the TZAP wt cells. How is this possible? What are the controls? Are the peaks normalized to the input sequence? To a non-specific antibody ChIP? (minor comment: the peaks in panel C seem unusually wide; the chromosomal loci of the peaks are not particularly of interest and could be given in a supplemental figure)

I think the authors could address this point w/o doing any exps.

MTFP1 regulation and telomere biology:

There are several problems with these experiments that need to be addressed before publication. First, the western blots on MTFP1 show no protein at all in absence of TZAP. But the mass spec shows that there is a 8fold reduction, which would lead to a visible MTFP1 band. Similarly, in the U2OS setting, RNAseq shows that the MTFP1 mRNA level is only 2fold down, yet the western blot shows no protein.

The authors should provide better western blots to address these discrepancies.

The analysis of mitochondrial status using mitotracker is underdeveloped since only two cells are shown (one wt and one KO). These two cells appear to be in different stages of the cell cycle, which could lead to different mitotracker patterns. Please provide a DIC images to show the whole cell and perform the analysis on at least 20 cells to make sure that cell cycle stages are equally represented in the KO and wt.

This should be easy.

Also, an shRNA control for the MTFP1 knockdown phenotype (in the wt HeLa clone) and a rescue of the KO cells with wt TZAP are needed to validate the observed results. Without these controls, the results could potentially be spurious.

I don't think these exps are needed.

In addition, In Tondera et al., observed sother phenotypes associated with MTFP1 KD such as cytochrome c release, increased apoptotic rate, reduced proliferation etc. These parameters (at least some) should also be checked.

If the authors have this data they can include in the sup data section but it's not critical.

Even with the additional controls and data, the link between telomere biology and mitochondrial biology is not well established and the text should be modified to include the possibility that TZAP simply has two unrelated roles, one in mitochondrial fusion and one in telomere biology.

This is possible, but I like the link between telomere biology and mito function-the authors should re-emphasize this point, and refer to the dePinho Nature paper again.

1st Revision - authors' response

20 March 2017

Please find attached the revised version of our manuscript "ZBTB48 is both a vertebrate telomere-binding protein and a transcriptional activator" (EMBOR-2017-44095-T).

Thank you very much for the quick and constructive review process. As you will see from the manuscript file and the point-by-point response, we have addressed all points raised by referee 1 and by referee 2 as suggested by referee 3 in the cross-comments. As a major point by referees 1 and 3, we have now included co-localization data for both endogenous ZBTB48 and overexpressed

FLAG-ZBTB48 in two telomerase-positive cell lines with long telomeres: HT1080 ST that were previously part of our study as well as HeLa 1.3 (as specified by the referees) that we kindly obtained on short notice from the de Lange lab. To address the related comment from referees 1 and 3 we have also revised our discussion to clarify that we see more robust co-localization in cells with long telomeres and that our study extends the finding of ZBTB48 as a telomere length regulator to cells with short telomeres.

We have also adapted the formatting changes as indicated in your decision letter and we have formatted our manuscript as a normal article including 6 main figures.

We are very thankful for the chance to revise our manuscript and hope that you agree that we addressed all comments conclusively. We believe that our manuscript is now ready for publication in *EMBO Reports*.

Author point-by-point response:

Referees 1 & 2: black

Referee 3: orange

Authors: blue

Referee #1:

Here the authors describe a similar effect of ZBTB48 at telomeres to that shown in a recent Science publication. They also reveal a novel finding of this protein as a transcriptional regulator. The manuscript is acceptable for publication but I would suggest the authors make a few small changes relating to the tone of the manuscript that in parts are over written - rather than being accurate and succinct. Here are some examples:

We appreciate the feedback and that this referee agrees that our manuscript is acceptable for publication.

1. In the introduction, authors state that large-scale screens had little overlap. The nature of these screens were quite different. In the context of other screens (siRNAs, CRISPR etc) the degree of overlap was actually comparably high. So the author's statement is misleading.

To avoid any confusion, we have removed the comment on the overlap between different screens for proteins associating with telomeres.

2. Authors should reference Li et al., 2017 directly after citing the name TZAP as it was this manuscript that initially described ZBTB48's role at telomeres and re-named it as TZAP.

We have added the citation directly after the TZAP name in the introduction as suggested.

3. Pg. 6. "Compared to a recent report..." When citing a difference with an existing observation the authors should state explicitly here whether the 2 studies utilized the same cell lines or not. It's not the same experiment otherwise.

We have clarified in this sentence that the authors have used the HeLa 1.2.11 clone to report frequent co-localization of exogenously overexpressed FLAG-ZBTB48 in telomerase-positive cells with long telomeres.

4. Figure 1 D-F. An enlarged merged insert would be helpful for visual clarity.

We have included enlarged merged inserts to improve visual clarity of the co-localization events.

5. Relating to Figure 2D-E. I would caution the authors from stating that there is no effect on U2OS cells based on this TRF analysis. U2OS telomeres should be resolved and quantified by PFGE as in Dilley et al., Nature 2016. Also, they should label the lanes 1, 2, 3, 4 etc.

We have labeled the lanes for individual clones as suggested and also included a visual aid to spot the average telomere lengths similar to Dilley et al. (PMID: 27760120). The teloblot for U2OS samples,

however, had already been resolved by PFGE. This is different from HeLa cells for which the short telomeres did not require pulsed-field resolution. Both experimental setups are described separately in the materials and methods section and we have now highlighted the PFGE aspect more clearly in the figure legend for Fig. 3B and in the text. As it remains possible that despite using PFGE, the resolution is insufficient to detect telomere changes relative to the average telomere length in U2OS cells, we cautiously state that “U2OS ZBTB48 KO clones did not show obvious changes in average telomere length.”

6. Figure 1C. Blots on right look terrible and highly processed. Should be replaced with better quality blots.

We have replaced the blots accordingly with less contrast. Please note that these blots are acquired as quantitative Western Blots on a digital Odyssey system (Li-Cor) with a different overall appearance compared to more conventional ECL staining recorded on film. In addition, FLAG-ZBTB48 ZnF11mut input levels are lower compared to the WT construct.

In addition, the authors declare and discuss in relative detail a minor conflict with the Li et al. 2016 regarding telomere length. This is based on observations with different HeLa clones. I would suggest that rather than make this an issue of contention, why don't they just go and test this in HeLa with long telomeres (HeLa 1.2.11 for example) and determine whether there is a greater association of TZAP with longer telomeres. The authors should do this. It's such a trivial experiment - and would be better for the field to resolve this here.

We have obtained the HeLa 1.3 clone from the de Lange lab and quantified the co-localization frequency of both endogenous ZBTB48 (using an antibody against ZBTB48) as well as exogenously overexpressed FLAG-ZBTB48 WT in HeLa Kyoto (our HeLa clone with short telomeres), HeLa 1.3 (long telomeres) and HT1080 super-telomerase cells as a second telomerase-positive cell line with long telomeres. With both the HT1080 super-telomerase and HeLa 1.3 cells, we can reproduce a frequent co-localization with telomeres upon overexpression even though the endogenous ZBTB48 levels at telomeres are significantly less frequent (Fig. EV1E-G).

We did not intend to state that our results contrast with the data presented by Li et al. (PMID: 28082411) as in general ZBTB48 is more easily detected at long telomeres, although this might be due in parts to technical reasons as even shelterin proteins are more easily detected at long telomeres. We mainly wanted to highlight that by showing that ZBTB48 also binds to short telomeres (Fig. 2C) and by observing a similar telomere-lengthening phenotype in cells with short telomeres (Fig. 3A), we are extending the previously described role in telomere length regulation. In other words, ZBTB48 seems to be an important telomere length regulator in different settings, involving both long (Li et al.) and short telomeres (our study), and it is not “only” limiting over-lengthening of already very long telomeres. We have adapted the paragraph in the discussion to make this clearer.

Referee #2:

In this interesting paper, the authors characterized the protein ZBTB48 (TZAP) that they previously identified as a telomeric protein. TZAP was recently described in a report in Science. Nonetheless, this clearly independent discovery of ZBTB48 as a telomeric protein is of great interest to the field. The authors first characterized the telomere binding property of the protein and demonstrated by mutation analysis that the last zinc finger domain (Zn11) is important for its *in vitro* binding to telomeric and subtelomeric sequences and its *in vivo* recruitment to telomere. They also suggested the potential function of TZAP in regulating the length of bulk telomeres although this reviewer is not entirely swayed by the data which is derived from clonal cell lines. These data are mostly consistent with the data published in the prior paper. The authors went on to characterize the non-telomeric, transcriptional function of TZAP. They found that the protein binds to various loci in the genome and most of the binding sites correspond to the promoter of genes. By RNAseq and MS, they show that various RNA and protein levels are changed after TZAP KO. Among them, they identified MTFP1, a mitochondrial protein, and showed that upon loss of TZAP, MTFP1 is depleted and some defects in mitochondria are reported. This latter part of the work merits more investigation. With suitable revisions, this manuscript would be excellent for publication in EMBO Report.

We would like to thank this reviewer for the appreciation of our work and the comments below.

Major comments:

Telomere length regulation/Figure 2D & 2E: To determine the role of TZAP in regulating telomere length, the authors do KOs of TZAP and look at telomere length in KO and wt clones. HeLa cells have a lot of clonal variation in telomere length, as the authors acknowledge, and as is shown in examination of their wt clones. Although the authors do a lot of work (analyzing many clones), this reviewer is very concerned that the results may be misleading. One out of five wt clones have long telomeres, whereas three out of five KO clones have long telomeres. Are we sure that if ten clones were examined for each, the numbers may not be much closer (say 4/10 and 5/10?). This is the inherent problem with analyzing telomere length in a clonal setting. To provide definitive evidence for the very important point that TZAP regulates telomere length in HeLa, the authors need to do a bulk population study over a number of PDs. This could be done with an shRNA to TZAP, or a bulk CRISPR KO (which now is possible). Without this data or other corroborating data (e.g. OE of TZAP showing telomere shortening), this important point is insufficiently supported by experimental evidence.

We agree with this reviewer that clonal variation is an important factor to be taken into account when determining telomere length in KO clones. At the same time, we agree with the feedback of referees 1 & 3 that we have systematically compared 5 WT and KO clones and that we could identify a significant increase of telomere length when comparing all clones. This finding is in line with the report by Li et al. (PMID: 28082411), consistently identifying ZBTB48 as a negative regulator of telomere length.

Transcriptional profiling/Figure 3:

This reviewer is very confused by the fact that the ChIP with TZAP on the KO U2OS or HeLa cells reveals more or less the same number of peaks as the TZAP wt cells. How is this possible? What are the controls? Are the peaks normalized to the input sequence? To a non-specific antibody ChIP? (Minor comment: the peaks in panel C seem unusually wide; the chromosomal loci of the peaks are not particularly of interest and could be given in a supplemental figure.)

To ensure that we only report binding sites that are highly specific to endogenous ZBTB48 we have ensured the following quality steps:

1. We have used the input DNA as control for peak calling for all samples (HeLa and U2OS WT and KO cells).
2. We performed all ChIP reactions with two independent ZBTB48 antibodies (Atlas and Genetex) and we used the same antibodies also on KO cells.
3. Using these KO controls we removed unspecific peaks from the WT.
4. For stringent reproducibility we have performed all reactions in technical duplicates, we have the two ZBTB48 antibodies as independent resources and both antibodies were ChIPed on two independent WT and KO clones in HeLa and U2OS. To clarify this point we have now re-labelled Fig. 3A/D to highlight the different antibodies on the WT and KO clones as true biological replicates and also updated the figure legends accordingly.
5. We only accept peaks for further analysis that were consistently found in all 4 WT samples (2 antibodies x 2 WT clones) with an at least 8-fold enrichment over the KO, a p-value < 0.01. In addition, we filtered out sites with a sequence coverage below rpk (reads per kilobase) < 100. While using KO cells is not commonplace yet, we would like to argue that using the same antibody and controlling precisely for individual background by removing the target protein is a more stringent background control compared to non-specific antibodies such as IgG. We would also like to highlight that the results reported here are highly consistent between both independent antibodies and they show a high degree of overlap between HeLa and U2OS samples. While Fig. 4B/E illustrate exact positions of individual peaks identified in U2OS and HeLa, respectively, Fig. 4C/F do not represent individual peaks. These panels illustrate the probability to call a peak at certain regions along chromosomes or in other words they represent a densitometry map of all peaks identified. The point of these panels is to illustrate if there are any preferences/biases towards the chromosomal location of ZBTB48 binding sites. Examples of individual ChIPseq peaks can be found in Fig. 5C and that those are ~500bp in diameter.

MTFP1 regulation and telomere biology: There are several problems with these experiments that need to be addressed before publication. First, the western blots on MTFP1 show no protein at all in absence of TZAP. But the mass spec shows that there is a 8fold reduction, which would lead to a visible MTFP1 band. Similarly, in the U2OS setting, RNAseq shows that the MTFP1 mRNA level is only 2fold down, yet the western blot shows no protein.

We would like to thank the referee for highlighting this point, which is inherent to many omics technologies. In order to be able to process entire datasets and to build ratios, label-free quantitative proteomics requires to impute zero values (= when no peptide was detected/quantified) (Cox et al., Mol Cell Proteomics, 2014; PMID: 24942700). Indeed, for all six replicates used for the proteomics analysis not a single MTFP1 peptide was measured whereas the six WT replicates show 1-4 peptides (up to 29% sequence coverage) (Dataset EV6). This is in agreement with the lack of detectable MTFP1 protein above the detection limit. Likewise, to model the RNAseq data we used DESeq2, which moderates the fold changes of genes with low counts (Love et al., Genome Biol, 2014; PMID: 25516281). We have now included a Maximum Likelihood Estimation of the “unshrunk” fold changes in Datasets EV4 and EV5. When inspecting the individual RNAseq tracks in Fig. 5C, it becomes clear that there are almost no detectable MTFP1 reads for both U2OS and HeLa ZBTB48 KO. This is slightly different for other factors such as PXMP2 for which we found some peptides in the HeLa ZBTB48 KO samples. Here, we would indeed expect at least a faint band by Western. To further validate this data, we have now included qPCR validation for all genes for which we had previously included genome browser tracks (Fig. 5D). In addition, we include an overexposed blot at the end of this point-by-point response to illustrate that the MTFP1 levels are simply below the detection limit.

The analysis of mitochondrial status using mitotracker is underdeveloped since only two cells are shown (one wt and one KO). These two cells appear to be in different stages of the cell cycle, which could lead to different mitotracker patterns. Please provide a DIC images to show the whole cell and perform the analysis on at least 20 cells to make sure that cell cycle stages are equally represented in the KO and wt. Also, an shRNA control for the MTFP1 knockdown phenotype (in the wt HeLa clone) and a rescue of the KO cells with wt TZAP are needed to validate the observed results. Without these controls, the results could potentially be spurious.

We have now included overview sample images, capturing each >10 cells each, to illustrate that the ZBTB48 KO cells recapitulate the previously described loss of MTFP1 phenotype for both HeLa and U2OS cells (Fig EV5).

In addition, In Tondera et al., observed other phenotypes associated with MTFP1 KD such as cytochrome c release, increased apoptotic rate, reduced proliferation etc. These parameters (at least some) should also be checked. Even with the additional controls and data, the link between telomere biology and mitochondrial biology is not well established and the text should be modified to include the possibility that TZAP simply has two unrelated roles, one in mitochondrial fusion and one in telomere biology.

We agree with the referee that those would be interesting additional parameters to test to further define the effect of MTFP1 depletion in the ZBTB48 KO background. As suggested by referee 3, we currently cannot provide these experiments. Whether the roles of ZBTB48 at telomeres and mitochondria are unrelated is somewhat semantic. Clearly, ZBTB48 impacts on both telomere and mitochondria biology which inherently makes it a link between both. Furthermore, given the binding both to telomeric DNA and the MTFP1 promoter, telomere length (at constant ZBTB48 expression levels) would likely impact on the amount of available ZBTB48 for transcriptional regulation. While we are in no way claiming that the presented data is sufficient to demonstrate a feedback loop between telomere length and transcriptional activity, it is conceivable based on our current knowledge of ZBTB48.

Minor comments:

Title and throughout: please refer to ZBTB48 and ZBTB48/TZAP so that the naïve reader does not get confused. Similarly for HOTT1, which has a second name used in the Dejardin paper that is cited, the second name should be given throughout. A search for HOTT1 would not yield anything other than the papers by this group.

We have included the multiple names used in different manuscripts, ZBTB48/HKR3/TZAP and HOT1/HMBOX1, in the abstract and we also mention these official aliases in the introduction to ensure that a pubmed search with any of these names would be successful. However, for better readability we stick to single gene names throughout the rest of the manuscript, which seems to be the common practise in most publications.

Figure 1B: The nomenclature of the mutants throughout the paper is confusing. FLAG-ZBTB49 ZnF11 sounds like a mutant with ZnF11 domain only instead of a complete protein with a single aa change in ZnF11. Please call it ZnF11mut or something similar.

We have clarified the naming of the mutant constructs accordingly throughout the manuscript.

Figure 1D: According to the Li et al. Science paper, after OE of TZAP in U2OS cells, there should be big blobs of TRF2 and TZAP, why is it not seen here?

We have not followed TRF2 and PML stainings upon overexpression of FLAG-ZBTB48 WT over time. Our early time points around 24h post transfection for the co-localization analysis are a possible explanation why we did not observe the “big blobs” seen by Li et al. (PMID: 28082411).

Figure 1D: Please indicate which antibody they used to visualize TZAP (anti-FLAG or TZAP). I suppose they used anti-FLAG. A western blot is needed to show expression levels of the different forms of TZAP.

We have used an antibody against endogenous ZBTB48 in Fig. 1D. The comparison with Fig. 1E, in which we used a FLAG-antibody to detect FLAG-ZBTB48 constructs, highlights lower (detectable) levels of co-localization on endogenous levels compared to the overexpression (~50% vs. ~80%). The expression of variant constructs of FLAGZBTB48 (WT, point mutants and deletion constructs) are shown as input samples in Fig 1B, Fig 1C, Fig EV1A and Fig EV1B. To avoid such confusion, we have changed labels in Fig. 1 and Fig. EV1. On the immunofluorescence images the antibodies are stated (ZBTB48 or FLAG) for endogenous stainings and overexpressions, respectively.

Figure S1: Please provide quantification for the experiments with HeLa and HT1080 ST cells and a western blot to compare the endogenous expression level of TZAP in the three cell lines tested.

We have now included a Western Blot showing the protein expression levels of ZBTB48 on endogenous level in HeLa Kyoto (short telomeres), HeLa 1.3 (long telomeres), HT1080 super-telomerase and U2OS in Fig EV1C.

Figure 2A & Figure S3: Why are the ChIPs quantified as a percentage of telomeric reads per total reads? This is not appropriate for ChIP. The read numbers should be normalized to the total telomeric reads in the input, not total reads in the ChIP. This is a standard approach for telomeric ChIPs.

The ChIPseq reads as shown in Fig. 2A and Fig. EV3 are represented as percentage of all reads containing at least 1 TTAGGG repeat (not the percentage of total reads). This representation has been previously used (Marzec et al., Cell, 2015; PMID: 25723166). We have now included the same representation shown for input samples in Fig. EV3. The specificity of the enrichment of “pure” telomeric reads with 7-8 TTAGGG repeats (within the 50bp sequencing reaction used in this study) is clearly seen by comparing TRF2, HOT1 and ZBTB48 antibody samples against input, IgG controls as well samples with the same antibodies used in HOT1 and ZBTB48 KO cells. While input samples are a widely used reference for normalization and peak calling not only for telomeric ChIPs but for ChIPseq in general, we would like to argue that the comparison to KO cells is inherently more rigorous as this controls for the entire workflow. Given that ChIP is a technology build on immunoprecipitation, we want to rigorously control for any potential biases within the IP procedure. For a regular protein IP, it is the default to compare to background binding controls such as IgG (or alternatively KO cells).

Figure 2C right panel: In order to compare the “enrichment” of TTAGGG in pulldown of two different cell lines, one has to normalize against the length of telomere. Otherwise, cells with longer telomeres will be likely to give higher enrichment because they have more telomeric DNA to be pulled down. If my understanding is correct, the 16x difference between HeLa and U2OS cells in

terms of TZAP binding is actually not significant considering that telomeres in U2OS they use is about 16x longer (Figure 2D & E). Hence, maybe the occupancy (or density) of the protein on telomere is the same between HeLa and U2OS. The reason that TZAP foci at telomere are not observed in HeLa cells may be that the telomeres are too short (fewer protein at telomere but with the same density).

We agree that telomere length is an important variable impacting relative binding. That is exactly why we have compared reads from anti-ZBTB48 ChIPseq reactions from ZBTB48 WT and KO clones. The absolute read counts shown in Fig. 2C indeed highlight a difference of almost two orders of magnitude ($2e3$ vs. $2e5$) between HeLa and U2OS samples. However, the background samples (ChIP reactions carried out in KO clones), are equally affected by such a difference in absolute number of read counts: For instance, the read counts in the KO clones shown in Fig. 2C are on average 336 in HeLa ZBTB48 KO, 186 in HeLa HOT1 KO and 191 for HeLa WT IgG samples in comparison to 7588 in U2OS ZBTB48 KO, 9891 in U2OS HOT1 KO and 5916 for U2OS WT IgG samples. Therefore, the relative TTAGGG enrichment in Fig. 2C (right panel) is experimentally normalised for differences in telomere length. Please note in addition that while ZBTB48 appears to be less densely distributed along HeLa telomeres (beyond the shorter telomeres compared to U2OS), the abundance of HOT1 seems proportional to the telomere length in HeLa and U2OS and TRF2 is relatively less abundant in U2OS cells (Fig. 2C), suggesting that those effects are not exclusively dependent on telomere length. While further analyses of this kind, involving more cell lines with different telomere lengths and telomerase-positive, telomerase-negative and ALT-positive backgrounds, should be carried out in the future, this data suggests that ZBTB48 binding to telomeres may not be directly proportional to the length of telomeres.

Referee #3:

Jahn et al documents additional TZAP functions not detailed by the authors' competitors in their recent Science paper. Importantly, these authors report that endogenous TZAP is present on short telomeres (as opposed to completely missing in the Science paper) and unappreciated roles as a transcription factor in mitochondria. Although its role in mitochondria is not well documented, this observation nevertheless could be important, given the previous observation by the DePinho lab linking critical telomere shortening to mitochondrial defects.

We would like to thank this referee for his/her constructive feedback.

Cross-comments from referee 1:

I understand the concerns of reviewer #2 which are well made. However, examining bulk telomere length over time also comes with problems (selection mainly). So, in my opinion, the authors have adhered to the emerging and correct methodology when dealing with CRISPR KO/shRNA variation (See Dilley et al., Nature 2016). In light of the fact that the great majority of the data presented in relation to telomere regulation are in line with the previous study, I don't think this is a major issue. It's okay to have some differences that can be reasoned for. For this reviewer's minor issues, these are mostly semantic/basic and can easily be fixed. For my own point, TZAP was reported as being the first de facto "counter" of telomere length, which is pretty important. So I think if the authors can show that they come to their conclusions by doing the same experiments as those shown in the Li et al. paper then I think everyone would be well served.

We agree with this referee that a telomere length counting molecule is a pretty attractive model. In general, our data is in agreement with the results presented by Li et al. (PMID: 28082411), e.g. ZBTB48 localization is more easily detected in cells with long telomeres etc. Whether ZBTB48 is actually sensing telomere length and functionally translating this information into telomere length control, is something we believe should be further investigated in quantitative detail in the future in a variety of settings (e.g. short/long telomeres & normal somatic/telomerase-positive/ALT-positive cells). Clearly, consistently between both studies ZBTB48 directly binds to telomeres and limits telomere elongation.

Cross-comments from referee 3:

Referee #1:

Here the authors describe a similar effect of ZBTB48 at telomeres to that shown in a recent Science publication. They also reveal a novel finding of this protein as a transcriptional regulator. The manuscript is acceptable for publication but I would suggest the authors make a few small changes relating to the tone of the manuscript that in parts are over written - rather than being accurate and succinct.

Here are some examples:

1. In the introduction, authors state that large-scale screens had little overlap. The nature of these screens was quite different. In the context of other screens (siRNAs, CRISPR, etc.) the degree of overlap was actually comparably high. So the author's statement is misleading.
2. Authors should reference Li et al., 2017 directly after citing the name TZAP as it was this manuscript that initially described ZBTB48's role at telomeres and re-named it as TZAP.
3. Pg. 6. "Compared to a recent report..." When citing a difference with an existing observation the authors should state explicitly here whether the 2 studies utilized the same cell lines or not. It's not the same experiment otherwise.
4. Figure 1 D-F. An enlarged merged insert would be helpful for visual clarity.
5. Relating to Figure 2D-E. I would caution the authors from stating that there is no effect on U2OS cells based on this TRF analysis. U2OS telomeres should be resolved and quantified by PFGE as in Dilley et al., Nature 2016. Also, they should label the lanes 1, 2, 3, 4 etc.
6. Figure 1C. Blots on right look terrible and highly processed. Should be replaced with better quality blots. In addition, the authors declare and discuss in relative detail a minor conflict with the Li et al. 2016 regarding telomere length. This is based on observations with different HeLa clones. I would suggest that rather than make this an issue of contention, why don't they just go and test this in HeLa with long telomeres (HeLa 1.2.11 for example) and determine whether there is a greater association of TZAP with longer telomeres. The authors should do this. It's such a trivial experiment - and would be better for the field to resolve this here.

I agree with Reviewer 1 that this experiment should be done – it also addresses rev 2's point 1.

Referee #2:

In this interesting paper, the authors characterized the protein ZBTB48 (TZAP) that they previously identified as a telomeric protein. TZAP was recently described in a report in Science. Nonetheless, this clearly independent discovery of ZBTB48 as a telomeric protein is of great interest to the field. The authors first characterized the telomere binding property of the protein and demonstrated by mutation analysis that the last zinc finger domain (Zn11) is important for its in vitro binding to telomeric and subtelomeric sequences and its in vivo recruitment to telomere. They also suggested the potential function of TZAP in regulating the length of bulk telomeres although this reviewer is not entirely swayed by the data which is derived from clonal cell lines. These data are mostly consistent with the data published in the prior paper. The authors went on to characterize the non-telomeric, transcriptional function of TZAP. They found that the protein binds to various loci in the genome and most of the binding sites correspond to the promoter of genes. By RNAseq and MS, they show that various RNA and protein levels are changed after TZAP KO. Among them, they identified MTFP1, a mitochondrial protein, and showed that upon loss of TZAP, MTFP1 is depleted and some defects in mitochondria are reported. This latter part of the work merits more investigation. With suitable revisions, this manuscript would be excellent for publication in EMBO Report.

Major comments:

Telomere length regulation/Figure 2D & 2E: To determine the role of TZAP in regulating telomere length, the authors do KOs of TZAP and look at telomere length in KO and wt clones. HeLa cells have a lot of clonal variation in telomere length, as the authors acknowledge, and as is shown in examination of their wt clones. Although the authors do a lot of work (analyzing many clones), this reviewer is very concerned that the results may be misleading. One out of five wt clones have long telomeres, whereas three out of five KO clones have long telomeres. Are we sure that if ten clones were examined for each, the numbers may not be much closer (say 4/10 and 5/10?). This is the inherent problem with analyzing telomere length in a clonal setting. To provide definitive evidence for the very important point that TZAP regulates telomere length in HeLa, the authors need to do a bulk population study over a number of PDs. This could be done with an shRNA to TZAP, or a bulk CRISPR KO (which now is possible). Without this data or other corroborating data (e.g. OE of

TZAP showing telomere shortening), this important point is insufficiently supported by experimental evidence.

See above-no need to do bulk telomere studies over time if they could use IF to document that TZAP localizes to long telomeres in HeLa1.2.11.

We now report frequent co-localization of overexpressed FLAG-ZBTB48 WT to telomeres in HeLa 1.3 cells and HT1080 super-telomerase cells with long telomeres with less frequent co-localization when quantifying endogenous ZBTB48. Please refer to the detailed answer to referee 1 above.

Transcriptional profiling/ Figure 3:

This reviewer is very confused by the fact that the ChIP with TZAP on the KO U2OS or HeLa cells reveals more or less the same number of peaks as the TZAP wt cells. How is this possible? What are the controls? Are the peaks normalized to the input sequence? To a non-specific antibody ChIP? (Minor comment: the peaks in panel C seem unusually wide; the chromosomal loci of the peaks are not particularly of interest and could be given in a supplemental figure.)

I think the authors could address this point w/o doing any exps.

We agree with this referee and have provided further explanation about how we control for only high confidence peaks above.

MTFP1 regulation and telomere biology There are several problems with these experiments that need to be addressed before publication. First, the western blots on MTFP1 show no protein at all in absence of TZAP. But the mass spec shows that there is a 8fold reduction, which would lead to a visible MTFP1 band. Similarly, in the U2OS setting, RNAseq shows that the MTFP1 mRNA level is only 2fold down, yet the western blot shows no protein.

The authors should provide better western blots to address these discrepancies.

Please refer to explanation about the analysis methods of the omics datasets above.

The analysis of mitochondrial status using mitotracker is underdeveloped since only two cells are shown (one wt and one KO). These two cells appear to be in different stages of the cell cycle, which could lead to different mitotracker patterns. Please provide a DIC images to show the whole cell and perform the analysis on at least 20 cells to make sure that cell cycle stages are equally represented in the KO and wt.

This should be easy.

As outlined above, we have now included overview images to represent the mitochondrial phenotype observed in ZBTB48 KO cells in both HeLa and U2OS.

Also, an shRNA control for the MTFP1 knockdown phenotype (in the wt HeLa clone) and a rescue of the KO cells with wt TZAP are needed to validate the observed results. Without these controls, the results could potentially be spurious.

I don't think these exps are needed.

We agree with the cross comment.

In addition, In Tondera et al., observed other phenotypes associated with MTFP1 KD such as cytochrome c release, increased apoptotic rate, reduced proliferation etc. These parameters (at least some) should also be checked.

If the authors have this data they can include in the sup data section but it's not critical.

We do not have this data at present and thank this referee for pointing out that this request is not critical to the study at this stage.

Even with the additional controls and data, the link between telomere biology and mitochondrial biology is not well established and the text should be modified to include the possibility that TZAP simply has two unrelated roles, one in mitochondrial fusion and one in telomere biology.

This is possible, but I like the link between telomere biology and mito function-the authors should re-emphasize this point, and refer to the dePinho Nature paper again.

We agree with this referee and we refer back to the dePinho Nature paper again in the discussion as suggested.

These blots are the same as in Fig 6A, with the addition of the overexposed MTFP1 blot to illustrate that MTFP1 is below detection limits in ZBTB48 KO cells.

2nd Editorial Decision

22 March 2017

Thank you for the submission of your revised manuscript. We have now received the enclosed report from referee 1 who was asked to assess it and who supports the publication of your manuscript. Only a few changes need to be made before we can proceed with its official acceptance.

Regarding statistics, it is not always clear to me whether "n" refers to individual cells of a single experiment, or to independently performed experiments (which it should). Can you please clarify this for figures 1, EV1 and 3?

In the manuscript text, please move the header "References" to just before the references, and change the header "Supplemental Figures" to "Expanded View Figures". We also need a running title and up to 5 keywords. Please also upload a manuscript word file. In the legend for figure EV1 the panels F and G are not mentioned and need to be added. Fig EV5 needs to be submitted in portrait format.

Please upload all movies as individual zip files that contain the movie and a movie legend file.

For our website, we also need a synopsis image, at the exact size of 550 pixels wide x 200-400 pixels high. The height is variable. The image can show either a model or key data.

We need ORCID IDs for all three corresponding authors. They need to insert them online in their profile page in our manuscript tracking system. We can unfortunately not do this for you.

I look forward to seeing a final version of your manuscript as soon as possible so that we can proceed with its rapid publication.

REFEREE REPORTS

Referee #1: Thank you for incorporating the suggested changes. Good job.

2nd Revision - authors' response

31 March 2017

Please find attached the revised version of our manuscript “ZBTB48 is both a vertebrate telomere-binding protein and a transcriptional activator” (EMBOR-2017-44095-V3).

We have now incorporated the last changes and corrections based on your recent feedback. We have added a running title and key words, moved the header “References” to the appropriate position, changed the header for supplementary material to “Expanded View Figures” and added figure legends to Fig. EV1F-G. The movie files are now given in a zip file with a separate movie legend file and the manuscript file is both provided in pdf and word format.

We have also clarified the statistics in Fig. 1, Fig. EV1 and Fig. 3. For Fig. 3 all experiments have been done with individual clones, which represent independent biological replicates, which we highlight by stating “independent (...) clones” in the figure legend. In case, you were referring to Fig. EV3B instead, those are indeed 3 biological replicate experiments for each of the 5 WT and KO clones (independent cell culture, independent extractions and independent qTRAP in technical triplicate reactions; 3x3 reactions each). We have thus kept “n = 3” in the figure legend. The quantifications of co-localization events in Fig. 1 and Fig. EV1 is referring to individual cells not 30-104 separate experiments. While this is the standard in the field, we agree that this was confusing and we now write e.g. “n = 102 cells” in the figure legend to make this more clear.

Finally, all corresponding authors have associated their ORCID IDs with the EMBO Reports account and we have included a synopsis image both at 550x343 pixels as well as in a larger size in case this might be useful.

3rd Editorial Decision

18 April 2017

I am very pleased to accept your manuscript for publication in the next available issue of EMBO reports. Thank you for your contribution to our journal.

YOU MUST COMPLETE ALL CELLS WITH A PINK BACKGROUND

Corresponding Author Name: Dennis Kappel
 Journal Submitted to: EMBO Reports
 Manuscript Number: EMBOR-2017-44095V2